

# Addressing deep array effects and impacts to wake steering with the cumulative-curl wake model

Christopher J. Bay[1], Paul Fleming[1], Bart Doekemeijer[1], Jennifer King[1], Matt Churchfield[1], and
Rafael Mudafort[1]

[1]National Wind Technology Center, National Renewable Energy Laboratory, Golden, CO, 80401, USA

**Correspondence:** Christopher Bay (Christopher.Bay@nrel.gov)

**Abstract.** Wind farm design and analysis heavily rely on computationally efficient engineering models that are evaluated many times to find an optimal solution. A recent article compared the state-of-the-art Gauss-curl hybrid (GCH) model to historical data of three offshore wind farms. Two points of model discrepancy were identified therein. The present article addresses those two concerns and presents the cumulative-curl (CC) model. Comparison of the CC model to high-fidelity simulation data and

historical data of three offshore wind farms confirms the improved accuracy of the CC model over the GCH model in situations with large wake losses and wake recovery over large interturbine distances. Additionally, the CC model performs comparably to the GCH model for single- and fewer-turbine wake interactions, which were already accurately modeled. Lastly, the CC model has been implemented in a vectorized form, greatly reducing the computation time for many wind conditions. The CC model now enables reliable simulation studies for both small and large offshore wind farms at a low computational cost,

thereby making it an ideal candidate for wake-steering optimization and layout optimization.

## 1   Introduction

Computationally efficient engineering models are critical for the design and analysis of wind farms. In the design of the wind farm layout or wind farm control strategy, a wake model is often evaluated many times. Such design processes require wind farm model evaluations to be completed in under a second for computational tractability, especially when considering

parametric uncertainty, design parameters, and control settings.

The development of such computationally efficient engineering models has been an active field of interest (e.g., Jensen, 1984). One recent model has been the FLOw Redirection and Induction in Steady State (FLORIS) model, developed by the U.S. National Renewable Energy Laboratory (NREL), the Delft University of Technology, and the University of Colorado Boulder. FLORIS is an open-source model that has widespread use in the literature: the design and analysis of wind farm

controls (e.g., Fleming et al., 2017, 2019; Doekemeijer et al., 2021; Campagnolo et al., 2020), layout optimization (e.g., Stanley, 2021), and coupled control/layout designs (e.g., Gebraad et al., 2017). FLORIS relies on the Gaussian wake model and the deflection model in Bastankhah and Porté-Agel (2016). Recent advances include incorporating the impact of curl into these models by the introduction of a pair of streamwise counterrotating vortices introduced by the yaw misalignment of wake steering (Martínez-Tossas et al., 2019; King et al., 2020). FLORIS has produced good agreement with smaller wind



farms in the literature (Doekemeijer et al., 2022). The focus of the FLORIS model has recently expanded to encompass the anticipated deployment of large offshore wind farms in Europe and the United States. The National Offshore Wind Research and Development Consortium (NOWRDC) include incorporating the impact advancing developments in FLORIS to enhance its validity for large offshore wind farms and to investigate the combined benefit of wind farm control and layout optimization for such wind farms at several U.S. offshore locations.

To date, two previous studies have investigated the accuracy of FLORIS with respect to historical supervisory control and data acquisition (SCADA) data of offshore wind farms. Hamilton et al. (2020) compared several combinations of wake and wake superposition models available within FLORIS to data from the Lillgrund Wind Plant. An important finding of that study was that accuracy in the leeward rear of the plant, where many upstream wakes impinge on a turbine, shows a strong sensitivity to the choice of the superposition model. They found that a linear wake superposition approach, rather than the commonly

used quadratic superposition approach, can reduce model error in this case. A second finding was that, given Lillgrund's close spacing, models that include near-wake modeling such as that of Blondel and Cathelain (2020) (a super-Gaussian wake model) reduce error relative to other common wake models. However, the authors did not find a single set of model choices that consistently improved model accuracy under all wind directions and wind speeds.

A more recent study by Doekemeijer et al. (2022) compared the current default Gaussian wake model in FLORIS v2.4

(described in a Sect. 2) to SCADA data from three offshore wind farms: Anholt, Offshore Windpark Egmond aan Zee (OWEZ), and Westermost Rough. The study found that the wake model in general performs well across the three wind farms when considering the case of one turbine waking another. These findings are different from Hamilton et al. (2020), likely because the turbines in the Lillgrund wind farm are spaced closely to one another. However, Doekemeijer et al. (2022) do confirm that the wind farm model underpredicts wake losses in the case of turbines in the rear of the wind farm, where many wakes overlap

and interact. Additionally, the article notes a possibility that wake losses for turbine pairs separated over a large distance (for example >25 rotor diameters, or >25$D$) are underpredicted.

The previously described situation in which many wakes interact and overlap, often far downstream in a large wind farm, is sometimes referred to as a "deep array" effect. Wake and wake combination models that behave well for small numbers of turbines tend to underforecast total wake losses at the back of large farms (Nygaard et al., 2020). There are various proposals in

the literature on how to correct for this discrepancy. Schlez and Neubert (2009) proposed the inclusion of a large-farm model correction to the ambient flow that is applied on top of the standard turbine wake models. The authors showed improved model agreement with SCADA data from the Horns Rev offshore wind farm. Further, Gunn et al. (2016) identified issues in the commonly used wake superposition methods, including the linear, sum-squared, and largest deficit approaches. The authors compared the wake-recovery engineering models to computational fluid dynamics (CFD) simulations and experiments; they

show good agreement in predicting the combined wake deficit in aligned two-turbine situations. However, under partial offset conditions, the combined wake is underpredicted by both wake superposition models, and instead, the authors suggested that a linear (deeper) wake superposition would be better. The inconsistency in which superposition model yields the best agreement with data matches the findings in Hamilton et al. (2020).



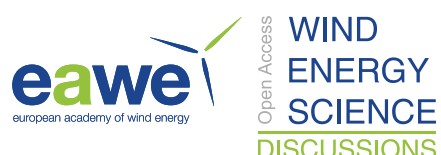

In this paper we present a combined wake deficit, superposition, and wake deflection model to accommodate the aforementioned discrepancies and yield accurate results for large offshore wind farms. This new model is a synthesis of several recently proposed models in the literature. We show that the model improves predictions of wake losses and wake control performance with respect to the Gaussian wind farm model (the default choice in FLORIS v2.4). The objective of this article is to explain and validate this new fast-running engineering wind farm model for wind farms of up to 100 wind turbines with respect to CFD and SCADA data of large offshore wind farms.

The organization of the paper is as follows: Sect. 2 presents the new engineering wind farm model. The revised wind farm model is compared to CFD simulation results in Sect. 3. Section 4 further validates the model by comparing it to the historical data of three offshore wind farms. The article is concluded in Sect. 5.

## 2  Engineering models

In this section we review both the state-of-the-art Gauss-curl hybrid (GCH) wind farm model (King et al., 2020) and the novel cumulative-curl (CC) wind farm model proposed in this paper. Both models are available in the FLORIS software framework (National Renewable Energy Laboratory, 2022a). The CC model builds on the GCH model. Section 2.1 presents the GCH model, after which Sect. 2.2 presents the CC model. Lastly, Sect. 2.3 explains how the parameters in the CC model are tuned.

### 2.1  The Gauss-curl hybrid model

The GCH model was first described in King et al. (2020). It was described as a combination of the Gaussian model detailed in Bastankhah and Porté-Agel (2014, 2016) and Niayifar and Porté-Agel (2015) with an approximation of the curl model of wake steering first presented in Martínez-Tossas et al. (2019).

The GCH model was compared to SCADA data in detail in Doekemeijer et al. (2022). Therein, the authors identified that the GCH model shows good agreement for smaller wind farms or wind farm subsets in terms of turbine wake effects and power production. Two areas for improvement were model superposition and farms with large distances between turbines. The CC model, described in the following sections, aims to resolve these model discrepancies.

### 2.2  The cumulative-curl model

The CC model combines the existing GCH model with improvements proposed in the literature. The cumulative-curl model builds on the GCH model by:

1. Replacing the current near-wake model in GCH with the model proposed by Blondel and Cathelain (2020): The near wake sets the trajectory for the far wake, which in turn enhances the accuracy of far-wake model predictions by slowing recovery in the far wake. This pattern fits with the change prescribed in Nygaard et al. (2020): "...the wake expansion is fastest closest to the turbine, where the wake contribution to the turbulence is largest. Further downstream the wake expansion slows down, asymptotically reaching a linear expansion at a constant rate".





2. Implementing the cumulative model of wind turbine wakes in wind farms proposed by Bastankhah et al. (2021) to replace the current Gaussian velocity deficit model and wake superposition submodels: The model proposed by Bastankhah et al. (2021) inherently calculates the combined wake effect rather than superimposing wakes as in the current GCH model, which showed limited accuracy in large wind farms (Doekemeijer et al., 2022). The CC model still contains the same deflection model from the GCH model as well as the secondary steering and yaw-added recovery effects driven by the counterrotating vortices generated by yaw steering.

3. Implementing the model into a vectorized structure, requiring careful considerations of the model equations to avoid the use of "for" loops over each wind direction/speed combination and taking advantage of single instruction, multiple data (SIMD): This approach is significantly faster than more traditional nested loops. The speed of analytical models is increasingly a pivotal factor for use cases such as wind farm design and online control. The implementation is included in the FLORIS v3 code, which is fully refactorized around vectorization and reduced memory overhead (National Renewable Energy Laboratory, 2022c). FLORIS v3 is open source and available at (National Renewable Energy Laboratory, 2022a).

The wake deficit takes the super-Gaussian form proposed in Blondel and Cathelain (2020), as shown in Eq. (1). The original cumulative wake model proposed in Bastankhah et al. (2021) does not include a near-wake model, which the super-Gaussian proposed by Blondel and Cathelain (2020) provides. The cumulative wake effect is included by adding the summation of $\sum_{i=1}^{n-1} \lambda_{ni} C_i / U_o$ into the wake center velocity deficit $C_n$ of the current turbine $n$, shown in Eq. (2). $\lambda_{ni}$ captures the wake contribution of upstream wind turbine $i$ on the value of $C_n$, as described in Bastankhah et al. (2021). Here, $\Delta u$ is the wake velocity deficit, $U_h$ is the average wind velocity inflow across the current turbine rotor, $U_o$ is the freestream velocity, $C_t, n$ is the thrust coefficient of the current turbine, $\gamma_n$ is the current turbine's yaw angle, $\Gamma$ is the gamma function, $\sigma_n$ is the wake half-width of the current turbine, $i$ is the index of the upstream turbines, $x$ are the downstream locations, $y$ are the lateral locations, $z$ are the vertical locations, $\delta y$ is the lateral wake deflection, **TI** is the turbulence intensity, and $a_f, b_f, c_f, a_s, b_s, c_{s1}$,



and $c_{s2}$ are tuned model parameters.

$$\frac{\Delta u}{U_h} = C_n e^{\left(\frac{-\tilde{r}^m}{2\sigma_n^2}\right)} \tag{1}$$

$$C_n = \left(1 - \sum_{i=1}^{n-1} \lambda_{ni} \frac{C_i}{U_o}\right) \left(a_1 - \sqrt{a_2 - \frac{mC_{t,n}cos(\gamma_n)}{16.0\Gamma\left(2/m\right)\sigma_n^{4/m}\left(1 - \sum_{i=1}^{n-1} \lambda_{ni}\frac{C_i}{U_o}\right)^2}}\right) \tag{2}$$

$$\tilde{r} = \frac{\sqrt{\left(y - y_i - \delta_y\right)^2 + \left(z - z_i\right)^2}}{D} \tag{3}$$

$$m = a_f e^{(b_f \tilde{x})} + c_f \tag{4}$$

$$a_1 = 2^{(2/m-1)} \tag{5}$$

$$a_2 = 2^{(4/m-2)} \tag{6}$$

$$\lambda_{ni} = \frac{\sigma_n^2}{\sigma_n^2 + \sigma_i^2} e^{-\frac{(y_n - y_i - \delta y)^2}{2(\sigma_n^2 + \sigma_i^2)}} e^{-\frac{(z_n - z_i)^2}{2(\sigma_n^2 + \sigma_i^2)}} \tag{7}$$

$$\sigma_n = k\tilde{x} + \epsilon \tag{8}$$

$$k = a_s \text{TI} + b_s \tag{9}$$

$$\tilde{x} = \frac{|x - x_n|}{D} \tag{10}$$

$$\epsilon = (c_{s1} C_t + c_{s2}) \cdot \sqrt{\beta} \tag{11}$$

$$\beta = 0.5 \cdot \frac{1.0 + \sqrt{1.0 - C_{t,n}}}{\sqrt{1.0 - C_{t,n}}} \tag{12}$$

The same deflection model is used from the GCH model, which includes secondary steering due to yawed wakes, and the yaw added recovery effects are also captured when updating the local turbulence intensity (TI) conditions for the current turbine, as defined in King et al. (2020).

### 2.3 Model parameters and tuning

In this article, we use the default model parameters for the GCH model in an approach similar to Doekemeijer et al. (2022). Because the CC model builds on top of the GCH model, the overlapping model parameters are unchanged. The model parameters new to the CC model are tuned to CFD data, of which the results are presented in Sect. 3.

In addition to the inherent model parameters, the wind farm model has two input parameters that have a dominating effect on the wake depth. The first input parameter is the TI of the ambient flow. TI has a linear effect on the wake spread and an inverse effect on the wake depth. Ideally, the TI in the CC model would be assigned according to an external physical measurement of the amount of turbulence in the flow, such as from a lidar or meteorological mast. However, literature suggests that TI may insufficiently represent the degree of wake recovery in the wind farm (Doekemeijer et al., 2020). Therefore, practice to date has been to select a data- and time-invariant value for the TI that, on average, best aligns with historical data. We continue that





practice in this article. More complex methods, such as partitioning by stability as in Ruisi and Bossanyi (2019), are reserved for future investigation and refinement.

The second input parameter is the wind direction variability, which defines the standard deviation of the probability distribution of the inflow wind direction over the averaging period. We classify this parameter as a tuning parameter that should capture measurement uncertainty, natural variations in the wind direction, the effect of time-averaging in the historical data, and the inherent slowness of the turbine yaw controllers. This approach is similar to those of Gaumond et al. (2013) and Doekemeijer et al. (2022), and we use a value of $3°$ as in Doekemeijer et al. (2022).

## 3   SOWFA analysis

Our analysis of the FLORIS improvements is a two-step process. In the first step, outlined in this section, the CC wind farm model from Sect. 2 is compared against wind farm flow data from a large-eddy simulation (LES) code called SOWFA (Simulator fOr Wind Farm Applications), developed at NREL (Churchfield et al., 2012b, a; Churchfield and Lee, 2014). Several model parameters specific to the CC model were tuned using data from these simulations. The second step will be a comparison of the CC model against SCADA data, to be presented in Sect. 4.

### 3.1   About SOWFA and configuration

SOWFA is a wind energy microscale flow solver that uses LES as its turbulence treatment technique such that the solver directly resolves the larger, energy-containing, turbulent flow scales and models the effects of the unresolved turbulent flow scales. LES is the highest-fidelity flow modeling technique currently feasible with today's high-performance computing systems for atmospheric flows. LES is a well-proven technique first applied to atmospheric flows nearly six decades ago by researchers

including Smagorinsky (1963) and Lilly (1962).

    SOWFA is built upon the popular, open-source, freely available OpenFOAM CFD toolkit(202, 2022). SOWFA solves the governing equations of fluid flow using the Boussinesq buoyancy approximation in which density is treated as constant everywhere but in the buoyancy term of the momentum equation where it is a function of virtual potential temperature (i.e., temperature with the effects of temperature change due to compression/expansion with altitude pressure change removed,

which is useful for understanding atmospheric stability). The effects of Earth's curvature and rotation are included through a Coriolis term. Surface momentum and heat fluxes are modeled using Monin–Obukhov scaling laws. Wind turbines are represented through actuator lines or disks (in this work, we only use disks), which apply body forces to the flow along lines approximating each blade or over the rotor-swept area, respectively. These body forces approximate the sectional aerodynamic in- and out-of-plane forces applied to the flow by the wind turbine blades. The actuator turbine models are coupled to the

NREL OpenFAST wind turbine structural-aerodynamics-servodynamic simulator (202).

    Simulations are run in two stages. In the first stage, often termed a "precursor", the computational domain (roughly 10 km × 5 km horizontally and 3 km tall with 10 m resolution within the boundary layer) is laterally periodic and the flow is allowed to cycle through the domain for many hours, during which a realistic turbulent atmospheric boundary layer forms and evolves.





In this stage, no turbines are simulated, and turbulent boundary data are collected during each time step (ranging from 0.3 s to 1 s, depending on flow speed) once the flow is fully developed (we typically ran the simulation for 6 hours before collecting 2 hours of boundary data). Precursor simulations represent different atmospheric conditions, which can be played through wind farm simulations.

In the second step, the domain is no longer fully laterally periodic. The inflow boundary condition is Dirichlet, using the saved time-varying turbulent boundary data, whereas the outflow side becomes an open boundary. The remaining two lateral sides remain periodic. Actuator disks are placed in the flow to produce wake effects. The first 20 minutes of these simulations are disregarded because wakes are propagating through the domain and are not fully developed. The problems solved in this work typically required the use of hundreds to more than 1,000 Intel Skylake compute cores of NREL's Hewlett-Packard high-performance computing system, Eagle.

This work was undertaken within the NOWRDC project "Wind Farm Control and Layout Optimization for U.S. Offshore Wind Farms", which seeks to investigate the value of wind farm control and layout optimization for potential U.S. offshore wind energy locations. Additionally, the project examines EU sites where historical SCADA data are available for model validation (to be considered in Sect. 4). With that in mind, we simulated U.S. sites in the Atlantic (Vineyard), the Pacific (Humboldt), and around Hawaii, and an EU North Sea site (Anholt).

To simulate specific days with specific conditions at these sites, we used SOWFA's mesoscale-microscale coupling capability. With that capability, we can influence the microscale using mesoscale (i.e, regional-scale) weather data. In this case, the mesoscale weather data for U.S. sites were provided from hindcasts performed using the Weather Research and Forecasting (WRF) mesoscale weather model developed by the National Center for Atmospheric Research. This coupling capability enhances the typical precursor step by providing a background driving force that drives the horizontally averaged LES velocity and potential temperature solution toward the vertical profiles provided by WRF at the location of interest (Allaerts et al., 2020). The resolved turbulence then naturally reacts to these time-evolving background mean conditions/forcings. Multiple WRF data sets covering both summer and winter were selected for each U.S. site and one of the EU sites (Anholt), and then LESs driven by each of those WRF data sets were performed. The simulations covered the full range of atmospheric stability and turbulence strength. For the EU site, similar data were obtained from the New European Wind Atlas (NEWA, https://map.neweuropeanwindatlas.eu/).

We also use WRF to provide annual wind roses and estimate turbulence intensity for each site, which is important for the comparison to SCADA data. WRF does not directly report turbulence intensity, but instead reports turbulent kinetic energy (TKE). TI is then estimated using the equation:

$$TI = \sqrt{TKE * 1.07}/ws \tag{13}$$

where TKE is turbulent kinenetic energy and ws is the wind speed.

Table 1 presents a subset of the 23 precursor simulations that were performed. In the table TI represents the ambient turbulence intensity as estimated via equation 13 using the TKE from WRF/NEWA. WD STD represents the standard deviation of the ambient wind direction impinging the wind farm in degrees. Note that wind direction is not exactly 270° (left to right) at the



**Table 1.** Summary of the SOWFA precursor simulations used in the full wind farm simulations.

| Precursor | Wind Farm | Wind Speed (m/s) | Wind Direction (deg) | TI (%) | WD STD (deg) |
|-----------|-----------|------------------|----------------------|--------|--------------|
| ah_3 | Anholt | 9.0 | 269.9 | 6.5% | 1.1 |
| ah_4 | Anholt | 8.6 | 269.8 | 6.5% | 1.4 |
| ah_5 | Anholt | 8.6 | 269.9 | 7.3% | 1.2 |
| ha_1 | Hawaii | 9.2 | 269.9 | 6.3% | 2.1 |
| ha_2 | Hawaii | 8.8 | 270.0 | 7.1% | 4.3 |
| hb_4 | Humboldt | 8.9 | 269.9 | 5.1% | 1.3 |

wind turbine hub height. From the original full set, this subset of six were selected for full wind farm simulation and analysis. The precursors that were omitted for analysis were removed due to issues caused by their high degree of atmospheric stability
(cases with strong negative surface heat flux). Two main issues were at play. The first issue is that the flow laminarized in some cases, producing turbulence levels much lower than those predicted by the corresponding WRF simulations. Laminarization of the flow under very stable conditions is a known LES challenge, and without real-world observational data, the LES results are questionable. Because WRF does not resolve any turbulence in the atmospheric boundary layer, but instead completely models it with a planetary boundary layer turbulence model, trust in WRF's turbulence predictions is also questionable. Therefore, we
did not include the cases with large turbulence disagreement between WRF and SOWFA. The second issue was that in some of the stable cases, atmospheric gravity waves formed. Atmospheric gravity waves are three-dimensional internal waves in the air caused by a vertical perturbation to stably stratified (in terms of density) flow. Wind turbines can be the disturbance that spawns these waves. In many of these cases, gravity waves formed above the wind farm, or sometimes were trapped within a near-surface, very stable layer. Although these may well be physically realistic effects, gravity waves reflect off of computational
domain boundaries and can pollute the flow with spurious reflections and amplifications. Our relatively low level of experience in properly treating these effects and the fact that these effects are beyond the scope of model improvements considered in this work led us to disregard cases involving atmospheric gravity waves. However, these cases also made us much more aware that gravity-wave effects on wind farms may be significant and should be studied in the future. Moreover, their effects should be considered as future additions to engineering wind farm flow models, including FLORIS. Thus, the remaining six precursor
simulations, while varying in turbulence level, represent only cases with a near-neutral to positive surface heat flux at the time of data sampling. It is important to note that nearly all of these cases are transient, and many, even though unstable at the time of interest, transitioned from a stable state earlier in the simulation. LES of the realistic, transient atmosphere is new to wind farm controls research.

The precursors were generated to control for the wind direction at a height of 120 m (the hub height of the International
Energy Agency Wind Technology Collaboration Programme 10 MW reference turbine). However, due to veer that occurs in the precursors, the wind direction can vary slightly at heights other than 120 m. The Wind Direction column in Table 1 captures this variation at 90 m (the hub height of the NREL 5 MW turbine), and is accounted for in the simulations.





Full wind farm simulations are run for a range of wind turbine yaw angles and lateral turbine locations, the latter to provide more realizations of the same flow for more converged flow statistics. In further analysis, the first 1,200 s of the simulation results are omitted because they relate to wake development and start-up effects. The remaining simulation output is time-averaged to obtain a steady-state representation of the wind turbine and farm performance. Using the time-averaged cubed wind speed field (as opposed to the cube of the time-averaged wind speed field) over the entire simulation domain, a virtual turbine can be placed at different locations downstream to estimate an expected power production if a turbine had been placed at that location. This approach was validated by comparing such a hypothetical turbine's power production to the power production of an actual turbine at the same location in the SOWFA simulation.

## 3.2 Single wake analysis

The first set of wind farm simulations in SOWFA analyze the wake of a single turbine. For each SOWFA precursor simulation in Table 1, wind farm simulations of a single NREL 5 MW reference turbine are run (Jonkman et al., 2009).

### 3.2.1 Wake recovery

As the accuracy of wind farm models is strongly connected to the accuracy of their wake-recovery submodel, this component is addressed first. Figure 1 compares the power production of a hypothetical downstream turbine at various distances behind an upstream turbine. In these simulations, the upstream turbine is aligned with the flow and thus wake steering is not yet considered. The TI assumed in the GCH and CC models was selected to yield perfect agreement with SOWFA at a $7D$ distance downstream. Proceeding downstream from $7D$, Fig. 1 shows that the wake recovery in the GCH model is overestimated with respect to SOWFA, while the new CC model shows much better agreement. This trend of slower wake recovery at large downstream distances is supported in literature (Nygaard et al., 2020).

The improvement of the CC model over the GCH model is largely largely attributable to the inclusion of the near-wake parameterization of the super-Gaussian model, based on the work of Blondel and Cathelain (2020). This parameterization yields a more gradual recovery following a shallower initial wake loss which results in an increased error in the near-wake zones in the $4D$–$7D$ range for the CC model in comparison to the GCH model. Figure 2 compares the absolute errors of the two models across the SOWFA simulations, confirming that the CC model shows a significantly lower error compared to the GCH model, notably for the far wake.

### 3.2.2 Wake steering of a single wake

Because wake steering is a key application of the CC model, an accurate representation of wake recovery under various yaw angles is essential. This subsection focuses on the differences in the effect of wake steering in the CC and GCH models. Figure 3 presents the power production of a hypothetical wind turbine at $7D$ downstream of a yawed upstream wind turbine. The upstream turbine is yawed for two different misalignment angles: -25° and +25°. Generally, the figure shows that the





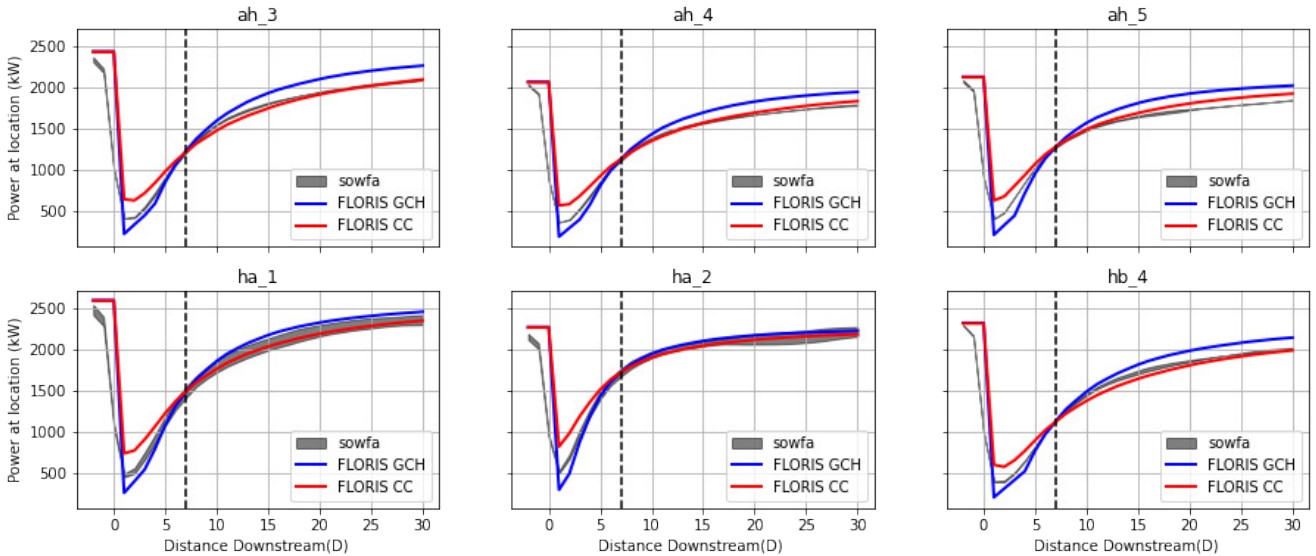

**Figure 1.** Power production of a downstream wind turbine located at various distances behind a yaw-aligned upstream wind turbine for each precursor described in Table1. The TI values for both the GCH and CC models were selected to match the wake depth in SOWFA at $7D$ distance (indicated via a dashed line). Note that empirical results have shown that the GCH model typically overpredicts recovery for distances $>10D$ (Doekemeijer et al., 2022).

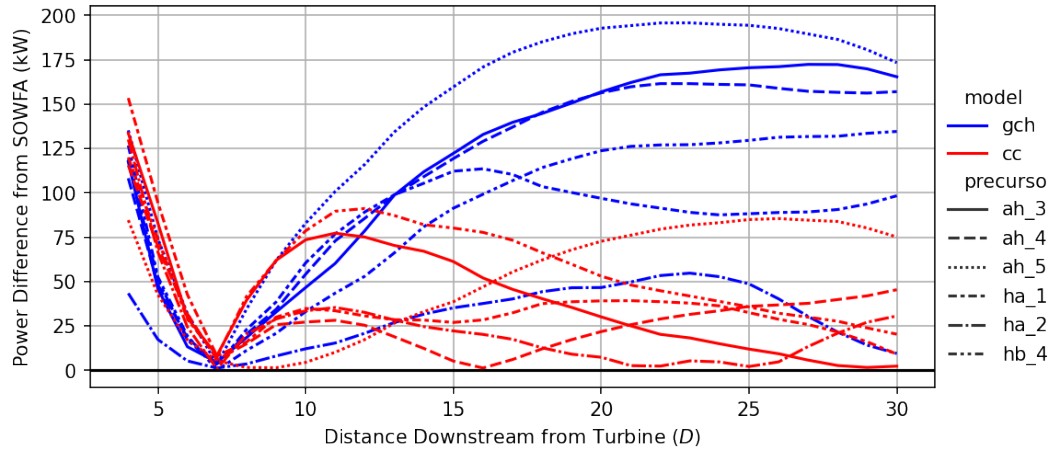

**Figure 2.** Absolute error between FLORIS and SOWFA predictions for the single-turbine wake-recovery characterization. The figure shows that the error has significantly decreased with the CC model compared to the GCH model, notably for the far wake.

difference between the CC and GCH models is small, which is reassuring because the GCH model has previously shown good agreement in wake-steering experiments (Fleming et al., 2021).



**Figure 3.** The power production of a hypothetical turbine 7 rotor diameters distance from an upstream wind turbine. The upstream wind turbine is misaligned by 25° in either direction. The left column of subplots compares a -25° (in the clockwise direction) change in yaw to the aligned case, and the right column displays a +25° yaw misalignment (in the counterclockwise direction). The difference between the CC and GCH models for single-turbine wake steering is small, which is reassuring because the GCH model has previously shown good agreement with experimental data (Fleming et al., 2021).





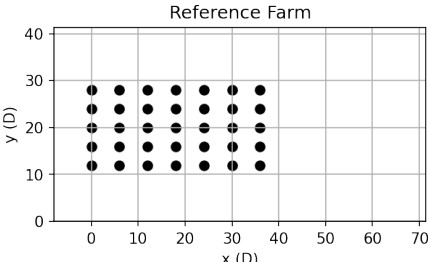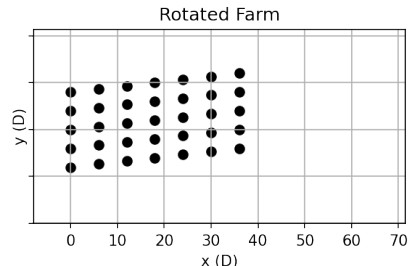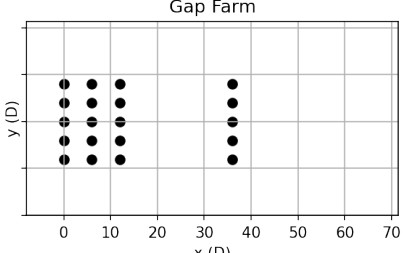

**Figure 4.** Wind turbine layouts of the wind farms used to validate the farm-scale effect in the CC model. Note that the units on the axes are turbine rotor diameters ($D$) and that axis dimensions match the simulation domain dimensions in SOWFA.

## 3.3 Farm-scale wake effects

Next, we compare the CC and GCH models to SOWFA simulations of larger, multi-array wind turbine farms. We consider three farm layouts, depicted in Fig. 4. The wind turbine model used in these simulations is the NREL 5 MW turbine from Jonkman et al. (2009). Each of the three wind farms depicted in Fig. 4 is organized into multiple rows, with each row containing five turbines spaced $4D$ apart in the cross-stream direction. The first wind farm is denoted "Reference Farm" and comprises seven rows, each spaced $6D$ apart in the streamwise direction. The second wind farm is denoted "Rotated Farm"; it differs from the first farm in that successive rows are shifted such that the columns of the wind farm are no longer aligned with the inflow direction. This layout is comparable to a situation in which the inflow wind direction creates partial wake overlap on downstream, waked wind turbines. The third wind farm is denoted "Gap Farm", and differs from the reference wind farm in that the fourth, fifth, and sixth rows are removed. The removal of rows creates a large gap in the farm to examine wake recovery over large distances, for which Doekemeijer et al. (2022) suggested that the GCH model may show model discrepancies in comparison to experimental data.

Figure 5 shows the row-by-row power production for each of the three wind farms and six precursor simulations, including the predicted values according to the CC and GCH models for each simulation, respectively. The top row represents the simulation results for the Reference Farm, the middle row represents simulations with the Rotated Farm, and the bottom row represents simulation results for the Gap Farm.

First, the simulations of the Reference Farm (Figure 5, top row) show excellent agreement between SOWFA and both the CC and GCH models. This confirms our notion that the GCH model is accurate for regular wind farms along columns of turbines. Note that these simulations exclusively represent simulations with full wake overlap (rather than partial wake overlap). For these simulations, all three models (SOWFA, CC, and GCH) show a pattern of significant reductions in the mean power production from the first to the second row of wind turbines, with the power production remaining fairly constant between the second and following rows.

Second, the simulations of the Rotated Farm (Fig. 5, middle row) show that the power production according to SOWFA between each wind farm row decreases as we progress further down the rows. This is in disagreement with the predictions of





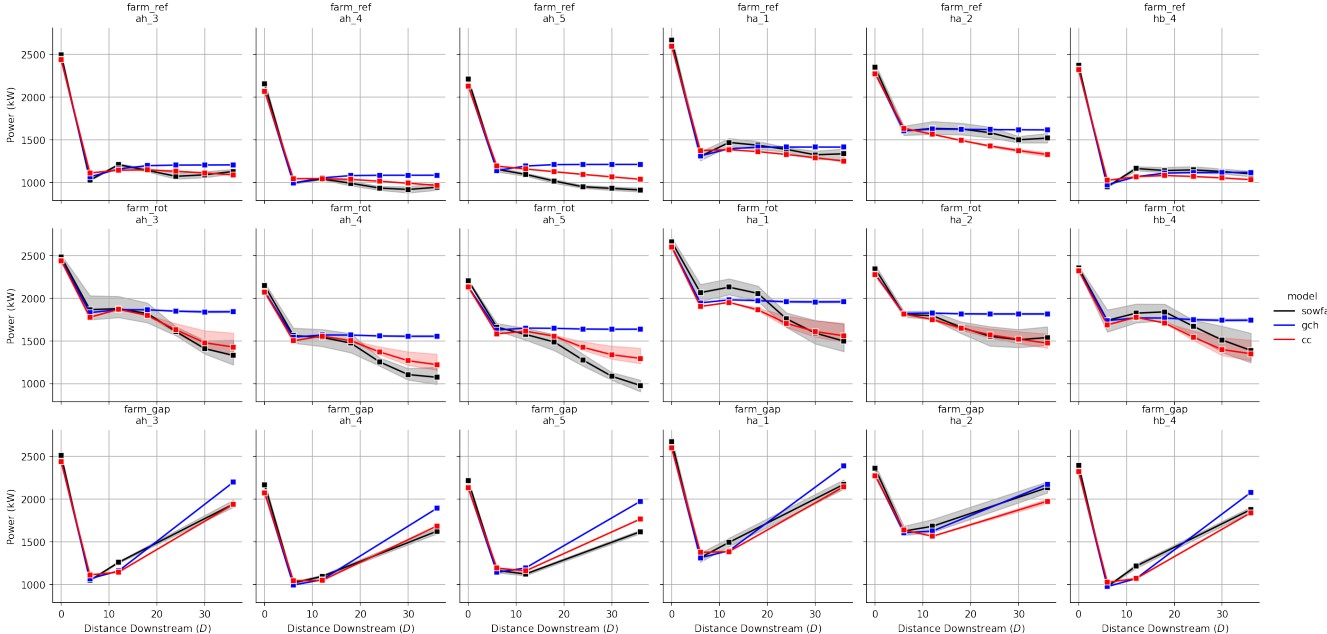

**Figure 5.** Comparing the power produced by row across farm types and models. Note that the plot points are average power for a row (with the shaded band indicating confidence interval), and the x-axis is the row's distance downstream from front turbines, in $D$. Each row of the plot is a farm type while each column is a precursor.

the GCH model, which shows a pattern of constant power production past the second row. On the other hand, the CC model
accurately captures the trend of decreasing power shown in SOWFA. This is because of the inherent accumulation of upstream
wake effects in the CC model, as opposed to the ad hoc wake superposition used in the GCH model. Accurately capturing this
effect is critical for wind farm applications, as this situation resembles the issue previously identified in Doekemeijer et al.
(2022). Therein, the authors showed a modeling mismatch between the GCH model and experimental data in situations where
the inflow wind direction was not aligned with the turbine arrays, yielding partial wake overlap.

Third, the simulations of the Gap Farm (Fig. 5, bottom row) show very similar results between the CC and GCH models
across the first three wind turbine rows. Note that these first three rows are equivalent to the first three rows in the Reference
Farm. Further downstream, however, there is a large gap of length $24D$ between the third and fourth row of turbines. Over this
gap, the GCH model shows a much higher wake recovery than SOWFA, which is consistent with the findings of Doekemeijer
et al. (2022) in comparing the GCH model with experimental data for such large gaps. The CC model resolves this issue and
295 more accurately captures the wake recovery over large distances. Nygaard et al. (2020) suggest that this pattern of recovery
is because wake recovery is initially driven by turbine-induced turbulence, causing significant wake recovery over relatively
short distances behind the rotor (up to $10D$). For larger distances, the dominant driver of wake recovery becomes ambient
turbulence, which leads to a much slower recovery of the wake. This effect was not previously addressed in the GCH model,
and has now been addressed in the CC model.





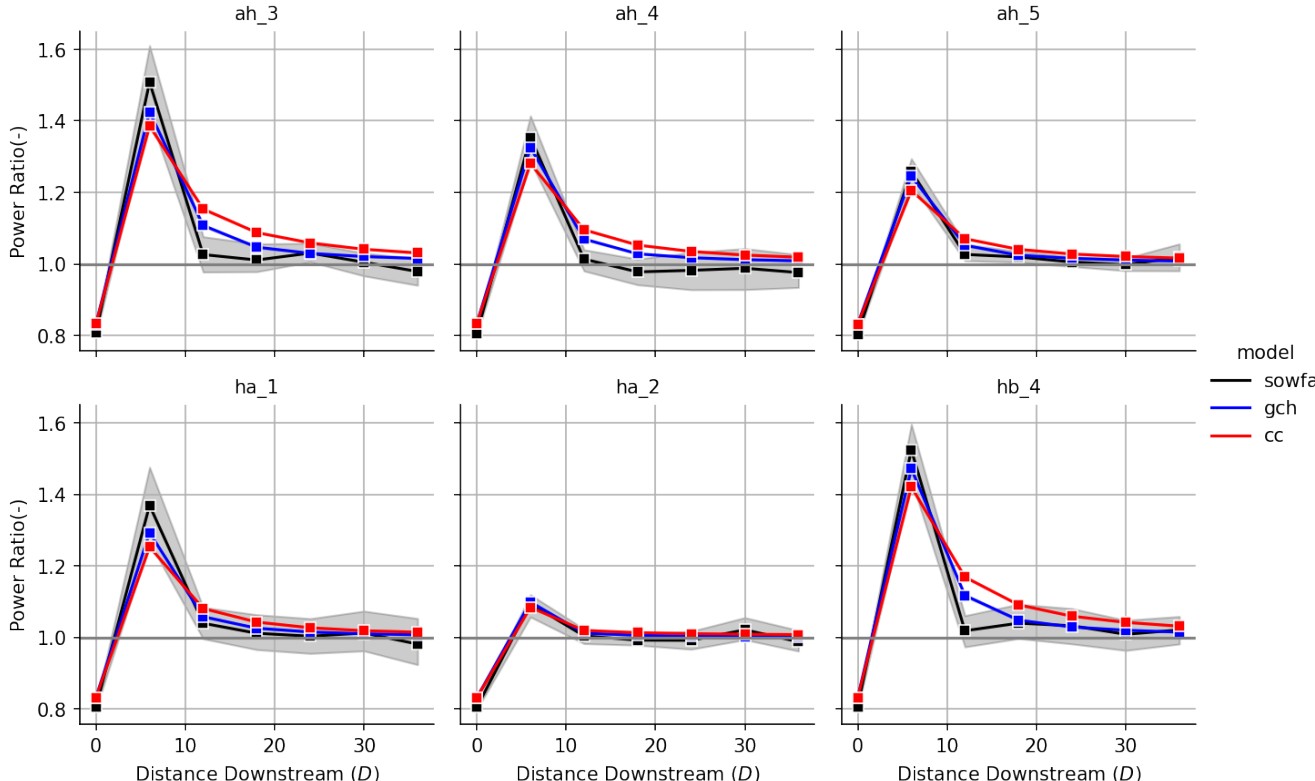

**Figure 6.** Comparing the ratio of the average power produced per wind turbine row (indicated via a point, shading indicates confidence interval) for the Reference Farm type across precursors when the front row of turbines is changed from aligned to a yaw of +25° (counterclockwise).

## 3.4 Farm-scale wake steering

The final suite of SOWFA simulations investigate the effect of wake steering on multi-array wind turbine farms. The wind farm layouts investigated are the same as those considered in the previous section, displayed in Fig. 4. The main purpose of these simulations is to compare the impact of wake steering for larger wind farms and in more realistic commercial applications.

First, Fig. 6 shows the the row-by-row wind turbine power production for the situation in which the turbines in the first row are yawed by +25° (in the counterclockwise direction). Again, the top row of plots refers to the Reference Farm, the middle row refers to the Rotated Farm, and the bottom row of plots refers to the Gap Farm. This figure shows that there is good agreement between SOWFA and both the GCH and CC models for the first two rows of wind turbines. Furthermore, both the GCH and CC models show a tendency to overestimate the impact of wake steering on the third row, yet the CC model outperforms the GCH model in all situations. The benefit of the CC model over the GCH model is most notable in the Reference Farm and the Gap Farm.




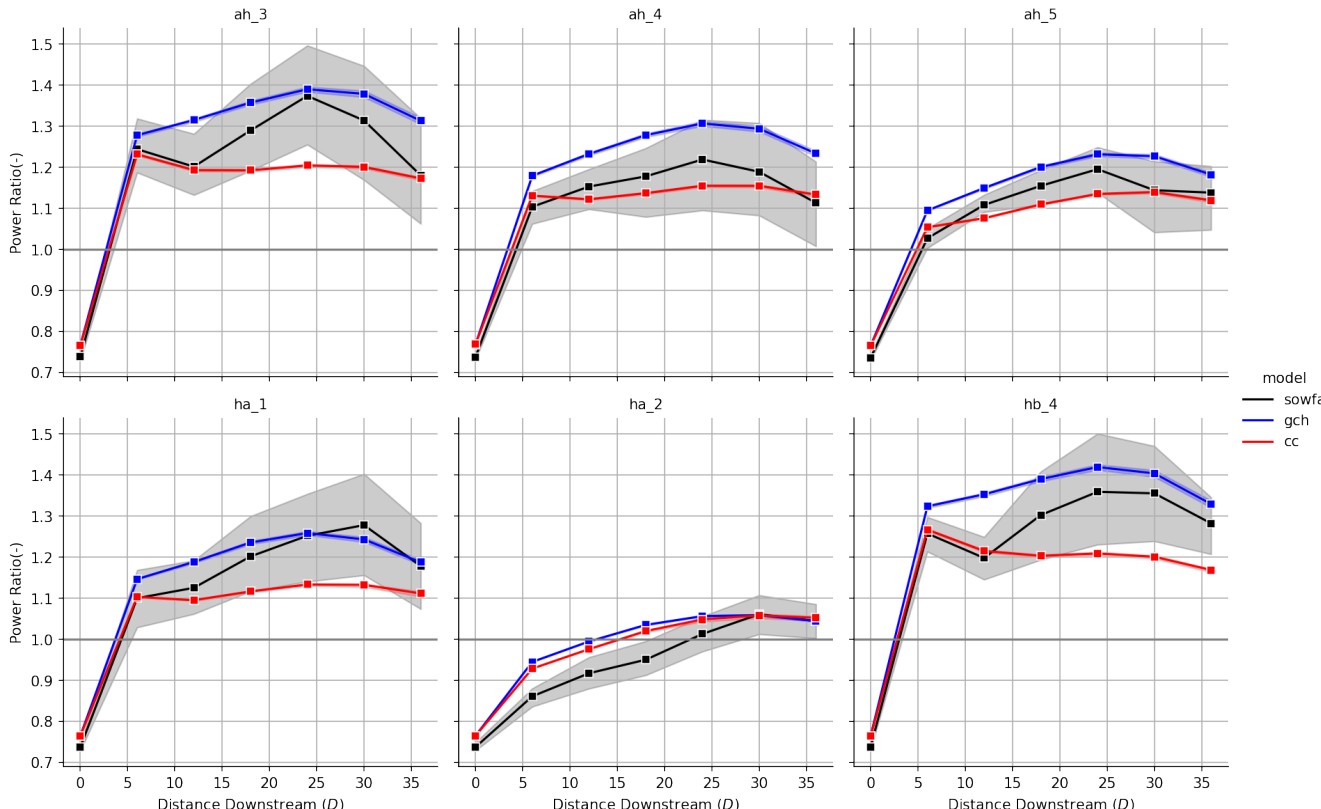

**Figure 7.** Comparing the ratio of the average power produced per wind turbine row (average indicated by the point, shaded band indicating confidence interval) for the Reference Farm type across precursors when applying a pattern of decreasing yaw angles per row.

Second, Fig. 7 presents the simulation results for the Reference Farm when a pattern of yaw angles is applied. This pattern assumes a large yaw misalignment angle for the first row of turbines, which then decreases linearly to zero for the last row of wind turbines. Due to high-computing facility resource limitations, these simulations were not performed for the Rotated and Gap farms. Figure 7 shows that the GCH model has a tendency to overestimate the wake losses at downstream turbines while the CC model more typically underestimates the wake losses. Also note that all models show a consistent increase in power production for all but the first row.

## 3.5 Reflection

Reflecting on the validation results in this section more closely, we find that both the GCH model and CC model do well for nominal and wake-steering operation in full wake overlap situations and regularly spaced wind farms. Furthermore, the CC model much better represents wake recovery under partial wake overlap and in irregularly spaced wind farms. when applying a pattern of decreasing yaw angles per row This will be considered in future work.





## 4 Model validation: comparison to historical SCADA data of three offshore wind farms

Now that fundamental components of the new CC model have been compared to high-fidelity simulation data in Sect. 3, the model is next compared to historical SCADA data.

### 325 4.1 Description of wind farms

The wind farms considered are three large offshore wind farms located in the North Sea in Europe: the Anholt, OWEZ, and Westermost Rough wind farms. The analysis in this article is predominantly a continuation of Doekemeijer et al. (2022), who compared the GCH model to the same historical data. The layouts of the three farms are illustrated in Fig. 8.

The historical data for the wind farms were provided by the wind farm owners – Ørsted and Shell. Each of the SCADA data
sets was postprocessed to remove measurement outliers and sensor-stuck faults. Furthermore, each turbine's nacelle heading was calibrated to true north by aligning the simulated energy ratios from FLORIS with the SCADA-based energy ratios. More details on the postprocessing can be found in Doekemeijer et al. (2022) and in the accompanying "FLORIS-based Analysis for SCADA data" (FLASC) software repository (National Renewable Energy Laboratory, 2022b).

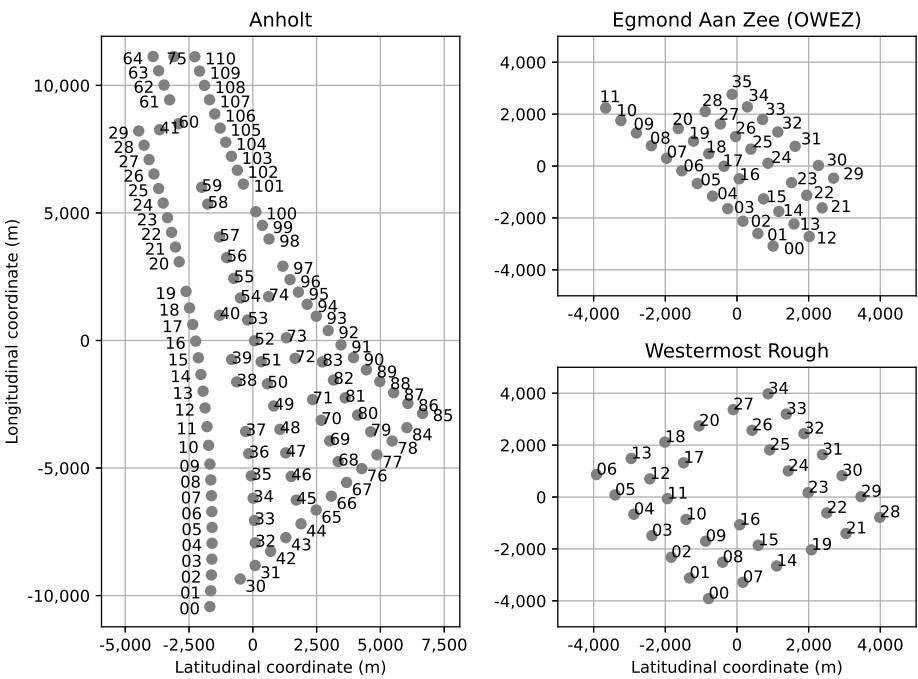

**Figure 8.** Layouts of the three offshore wind farms of which the SCADA data are compared to the CC and GCH wind farm models. Illustration taken from Doekemeijer et al. (2022) with permission.





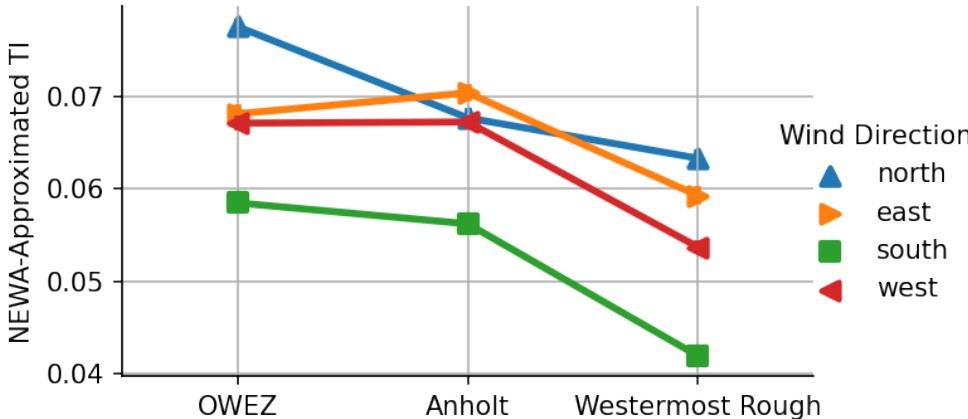

**Figure 9.** Average ambient turbulence intensity (TI) based on New European Wind Atlas (https://map.neweuropeanwindatlas.eu/) mesoscale simulations for the three European offshore wind farms. Note that TI is approximated from turbulent kinetic energy using Eq.13

Further, Doekemeijer et al. (2022) assumed a fixed value for the ambient turbulence intensity of 6% and an ambient wind direction standard deviation of 3° in their analysis. In this article, we follow the same assumption for the ambient wind direction standard deviation but instead base the values for the ambient turbulence intensity on the WRF simulations described in Sect 3. The ambient turbulence intensity values for each site are computed for four wind direction sectors, as notable differences can be expected from coastal effects and neighboring farms. The estimated turbulence intensities from WRF for each farm are displayed in Fig. 9.

Out of the three offshore wind farms, the Westermost Rough wind farm shows the lowest ambient turbulence according to WRF. For all three farms, winds from the south are expected to be paired with the lowest ambient turbulence intensity.

Finally, Doekemeijer et al. (2022) largely relies on energy ratio as a validation metric. The energy ratio represents the power production of a turbine in comparison to an unwaked turbine, and thereby represents the degree of wake losses for a single wind turbine. For a more exact definition of the energy ratio, see Doekemeijer et al. (2022). Similar to Doekemeijer et al. (2022), for the analysis in this article we reduce the data set to ambient wind speeds of 6–10 m/s because wake losses are most prominent in this range, thus yielding the most informative energy ratio curves.

## 4.2 Single-array wind farm analysis

In this paper we reapply the analysis methods described in Doekemeijer et al. (2022). One primary analysis method therein is to calculate the energy ratios of an aligned array of a subset of wind turbines in the wind farm. In this single-array analysis, the energy ratio of each turbine in the array relative to the upstream-most turbine for a narrow sector of wind directions (±7.5° around the wind direction that causes full wake overlap, i.e., perfectly aligns the turbine array) is calculated and compared. The energy ratio curves for four different turbine arrays are presented in Fig. 10.

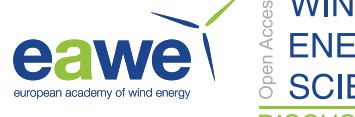

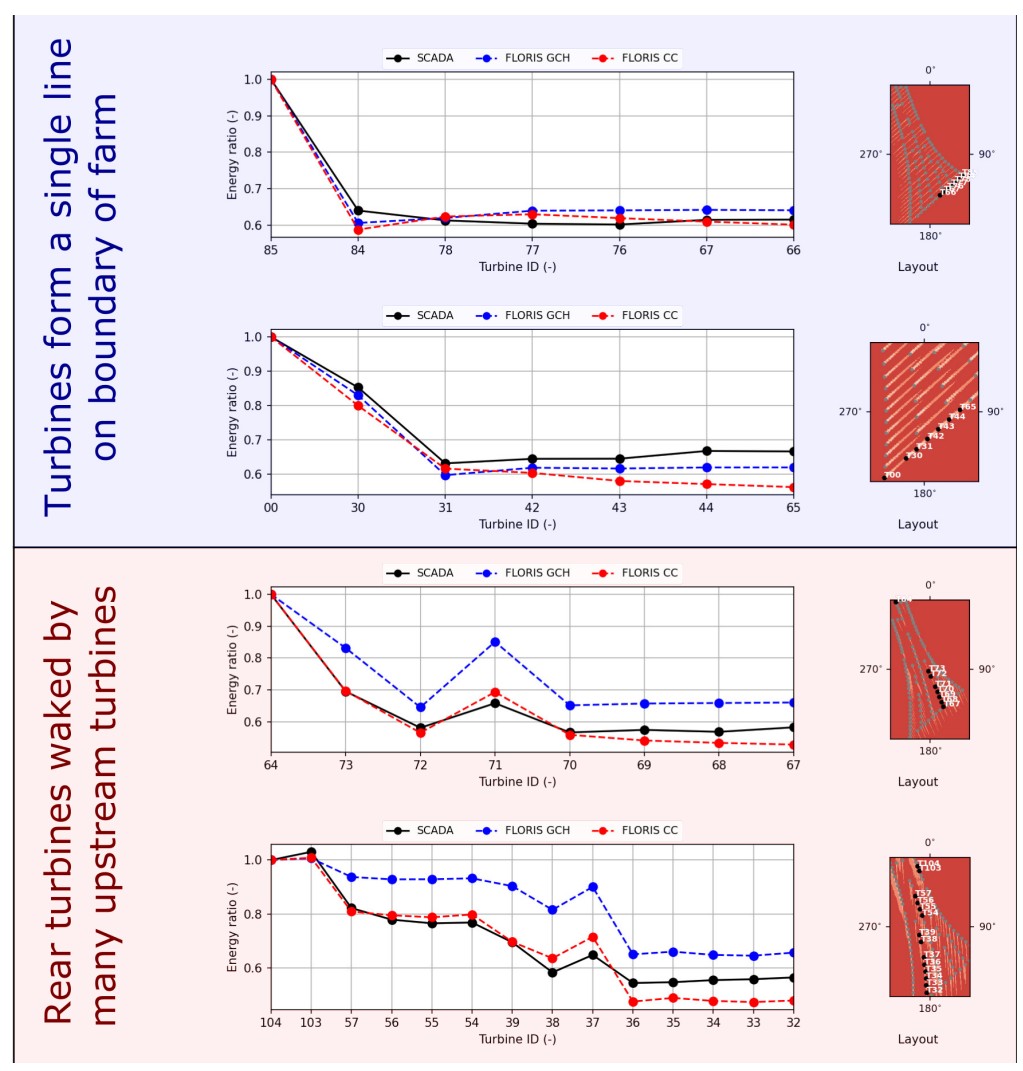

**Figure 10.** Ratio of energy produced by wind turbines in a row with respect to the energy produced by the first turbine in the row. The energy ratios are normalized to the most upstream wind turbine in the row. The wind direction used to bin the data is assumed to be equal to the nacelle heading of the most upstream wind turbine. The four turbine arrays considered are all from the Anholt wind farm, which is an excellent candidate for these validation studies because of its sheer size (111 wind turbines).





Through such turbine-array analyses, Doekemeijer et al. (2022) found that the GCH model agrees well with the SCADA data for turbine arrays on the perimeter of the wind farms. The GCH model showed larger divergence for turbine arrays centrally 355 positioned in the wind farm, whereas the CC model is expected to improve matching in these cases. This is confirmed by Fig. 10. The CC model shows deeper losses for turbines central to the Anholt wind farm (bottom two subplots) while not performing notably worse for turbines on the outer perimeters of the Anholt wind farm (top two subplots).

While insightful, this single-array analysis approach has relatively large uncertainty bounds. The energy ratios are calculated based on a single reference turbine, the most upstream wind turbine. Additionally, it only considers a 15° wind direction sector 360 to calculate a single energy ratio for every wind turbine, but there may be significant variations in the turbine's power production across different wind directions in this sector. Therefore, we consider a second methodology in alignment with the approach in Doekemeijer et al. (2022).

### 4.3 Wind-farm-wide analysis

A secondary analysis method to compare model predictions with SCADA data is done by calculating the energy ratio of a 365 single turbine across the entire wind rose. The energy ratio curves across the entire wind rose are computed using the same methods as Doekemeijer et al. (2022). The energy ratio of this turbine is now normalized to all turbines operating in freestream (and is thus wind-direction-dependent) within a 5 km radius for the OWEZ and Westermost Rough wind farms. Due to the size of the Anholt wind farm, the freestream turbines are defined as the five freestream-operating turbines closest to the turbine for which we are calculating the energy ratio. The use of multiple reference turbines significantly reduces the uncertainty of 370 the found energy ratios and generally provides energy ratio curves more in line with the model predictions. Additionally, these energy ratios across the wind rose are calculated for two different wind direction binning widths: 3° and 30°. The two bin sizes have different advantages. The energy ratios calculated using 3° binning provide a higher resolution of the wake effects, and therefore single-wake profiles are more visible. The disadvantage of such a narrow bin width is that it increases the sensitivity of the energy ratio to measurement noise and specific model choices (i.e., the choice of standard deviation on the inflow wind 375 direction, which is 3° as discussed in Sect. 4.1). The energy ratios with 30° binning represent a wider-perspective energy ratio (i.e., generalized wake loss for a larger wind direction sector), which is essentially insensitive to the choice of wind direction variability and less sensitive to the underlying uncertainty in wind direction measurements.

The energy ratios across the wind rose are calculated for a set of handpicked wind turbines, similar to Doekemeijer et al. (2022). The wind turbine cases shown are primarily selected via the following criteria:

– The turbine of interest and several neighboring turbines have a consistent northing calibration throughout the entire data set (i.e., the definition of true north does not change). This means their wind direction measurements can be used to derive which turbines are in freestream.

 – The turbine is in a relative position such that it is sometimes behind a single turbine, or single column, and at other times in the wake of a larger cluster of turbines. This provides a sufficient diversity in the energy ratio curve to validate the CC 385 model over a wide range of wake scenarios and operating conditions.



- The energy ratio curves do not include values greater than 1. A value higher than 1 likely indicates artefacts in the turbine performance curves or points toward another oddity in the comparison. In Doekemeijer et al. (2022), a part of these energy ratios were shown to be due to heterogeneity in the ambient wind speed; however, the consideration of heterogeneous effects is outside the scope of this work.

### 4.3.1 The Westermost Rough offshore wind farm

The first analysis concerns the Westermost Rough offshore wind farm. This wind farm is expected to experience the lowest ambient turbulence and therefore higher wake losses (Fig. 9). Westermost Rough is also unique in that it features a gap in the center of the farm, similar to the Gap Farm discussed in Sect. 3. It therefore makes an excellent candidate to validate the CC model for large interturbine spacing.

The energy ratio curves of Turbines 17 and 14 across the entire wind rose and their relative position in the farm are presented in Figs. 11 and 12. These turbines are positioned on the north and south side of the farm, respectively. The agreement of the CC and GCH models with the SCADA data is shown. For the directions in which the wake losses are from a single or small number of turbines, the GCH and CC models predict near identical energy ratios, and both agree very well with the historical data. The GCH and CC models diverge for wind directions in which many, nonaligned turbines are upstream of the considered

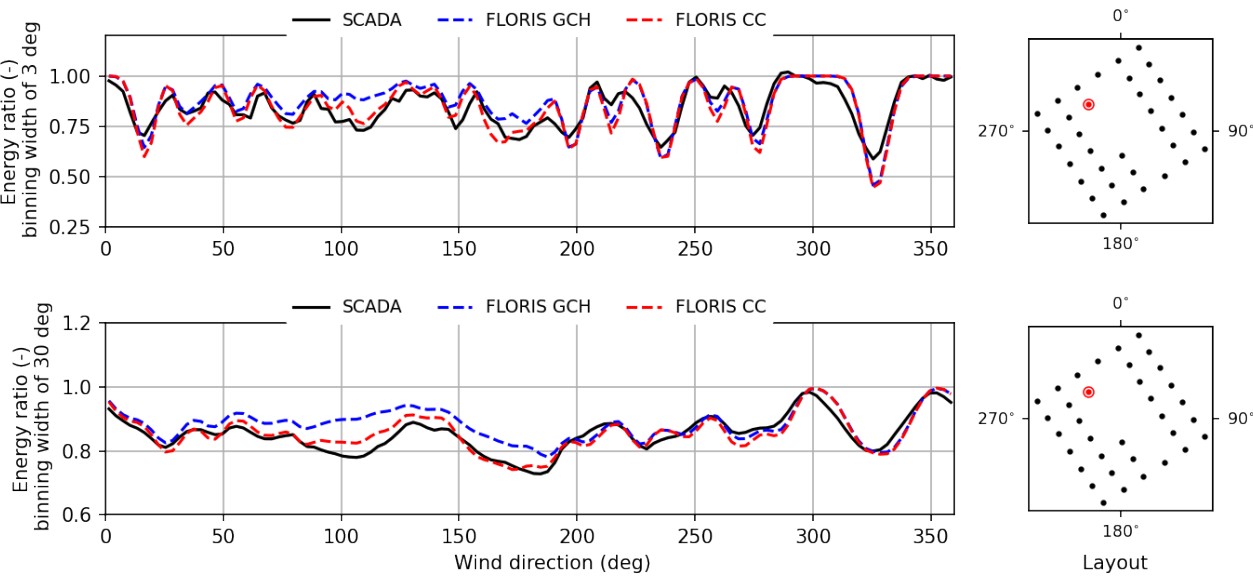

**Figure 11.** Energy ratio versus wind direction for Turbine 17, located in the northwest of the Westermost Rough wind farm. The reference turbines to which the energy ratio is normalized are the turbines that are both operating in freestream flow and within a 5 km radius of Turbine 17. The reference wind direction is determined by the nacelle headings from turbines directly neighboring Turbine 17.

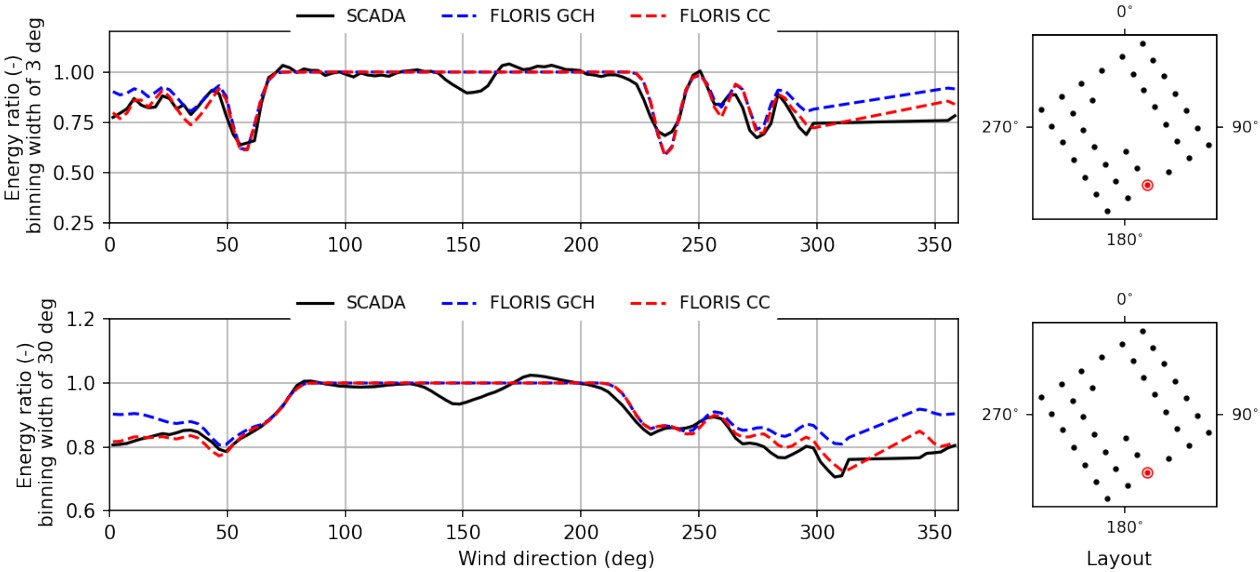

**Figure 12.** Energy ratio versus wind direction for Turbine 14, located in the south of the Westermost Rough wind farm. The reference turbines to which the energy ratio is normalized are the turbines that are both operating in freestream flow and within a 5 km radius of Turbine 14. The reference wind direction is determined by the nacelle headings from turbines directly neighboring Turbine 14.

turbine (southeast for the turbine in Fig. 11 and north for the turbine in Fig. 12). The CC model significantly outperforms the GCH model for these deeper-wake-loss situations, and generally agrees very well with the historical data.

### 4.3.2 The Anholt offshore wind farm

The second wind farm considered is the Anholt offshore wind farm off the coast of Denmark. With 111 turbines it is by far the largest wind farm of the three considered in this article. The size of this wind farm makes it an excellent condition to validate
the deep array effects identified in Sect. 3. In the simulation studies of Sect. 3, it was suggested that turbines on the perimeter of the farm are modeled accurately in the GCH model but that the wake recovery is overestimated for turbines central in a farm. This notion is now further assessed using the historical data of the Anholt offshore wind farm.

Figure 13 presents the energy ratio curve across the wind rose for Turbine 86, located in the southeast corner of the farm. An energy ratio curve pattern similar to what was observed for Westermost Rough can be viewed. At a wind direction of
120°, Turbine 86 is waked by a single neighboring turbine and the GCH and CC models perform equivalently well. Both yield accurate predictions compared to the SCADA data. Further, for a direction such as 310°, where the turbine is located far downstream near the rear of the large farm, the energy ratios for the CC model outperform those for the GCH model, notably



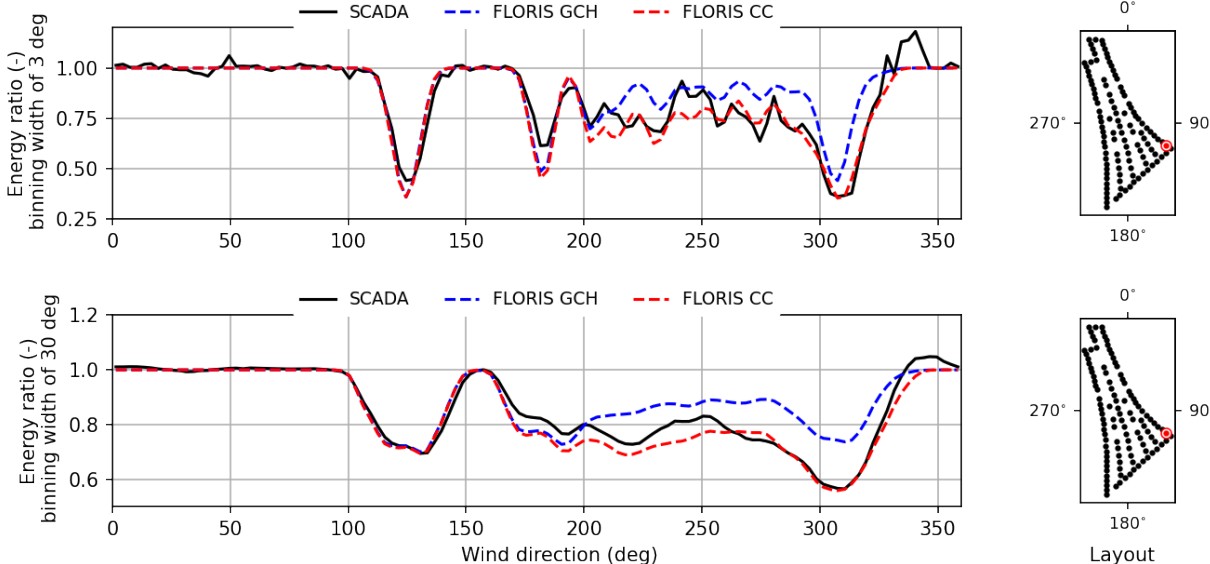

**Figure 13.** Energy ratio versus wind direction for a turbine located in the southeast region of the Anholt wind farm. The reference turbines to which the energy ratio is normalized are the five closest wind turbines that are operating in freestream flow. The reference wind direction is determined by the nacelle headings from turbines directly neighboring Turbine 86.

for the 30° binning. The improvement is stark: the energy ratio of 0.58 for the CC model is much closer to the observed SCADA value versus the energy ratio of 0.75 predicted by the GCH model.

Figure 14 represents the energy ratio curve for Turbine 51, located in the center of the farm. Important to note is that for wind coming from the west and east, the number of upstream turbines is smaller, so conditions are more like those for smaller wind farms. The GCH and CC models show similar energy ratios for the wake losses in the direction of 60° and 270°, where the farm is "thin" and there are relatively few wake interactions. The two models significantly diverge for wind directions near 0° and 180°, in which there is significant wake overlap and large arrays of turbines waking one another. Observing Fig. 14, it

can be said that in general the CC model outperforms the GCH model, notably in situations with deeper wake losses.

### 4.3.3   The OWEZ offshore wind farm

The third and final wind farm considered is the OWEZ wind farm, located off the coast of the Netherlands. This wind farm is older, and the historical data typically show more measurement outliers and erratic behavior. However, since it has a very representative layout, it is an excellent candidate for a generalized evaluation of the model. The energy ratio curve for Turbine

22 is presented in Fig. 15. In this figure, we again find excellent agreement with the SCADA data for single-turbine wake



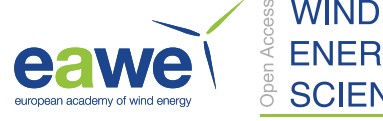

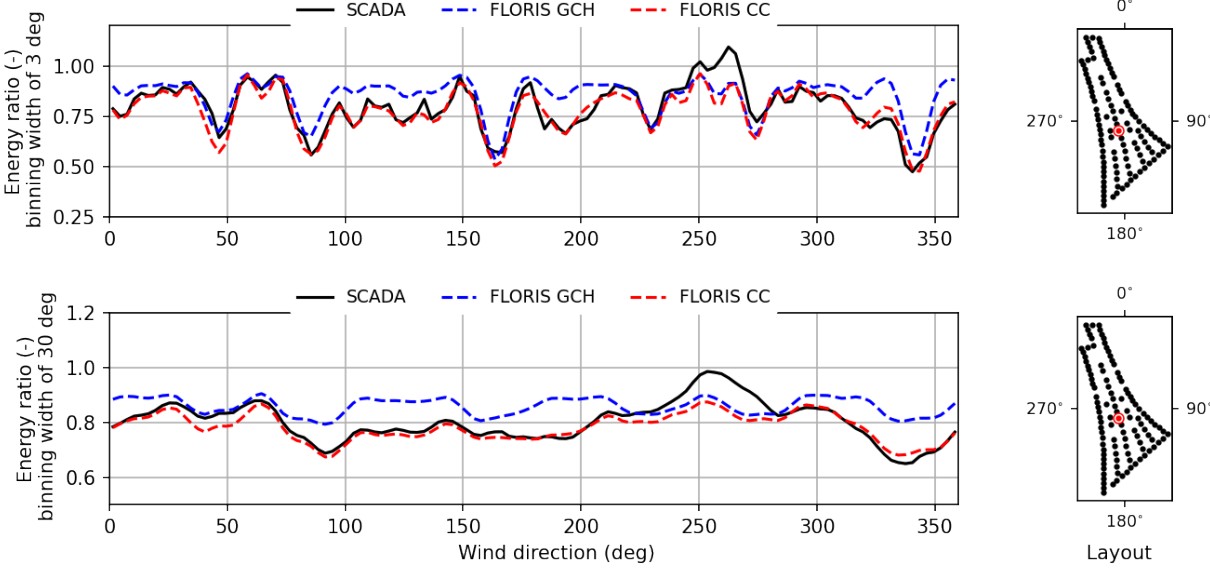

**Figure 14.** Energy ratio versus wind direction for Turbine 86, located in the center of the Anholt wind farm. The reference turbines to which the energy ratio is normalized are the five closest wind turbines that are operating in freestream flow. The reference wind direction is determined by the nacelle headings from turbines directly neighboring Turbine 86.

interactions for both the CC and GCH models, and the two models hardly differ from one another. Further, for multiple-turbine wake interactions when winds come from the northwest, deep-array-type conditions become apparent. For this direction, the CC model shows better agreement than the GCH model with the historical data. Interesting to note is the situation at a wind direction of 320°. Here, Turbines 27 through 21 perfectly align and cause full wake overlap. This is especially clear for the

energy ratios with 3° binning. The simulation results from Sect. 3 suggested that both the GCH and CC models should yield accurate predictions for these situations, which is confirmed by Fig. 15.

## 4.4 Reflection

The results in this section demonstrate that the CC model performs comparably to the GCH wind farm model under single- and few-turbine wake interactions and significantly outperforms the GCH model for effects commonly seen in larger wind farms.

The two effects specifically addressed through the CC model are 1) larger wake losses deep in a wind farm and 2) deeper and more persistent wake losses over longer distances behind a wind turbine.

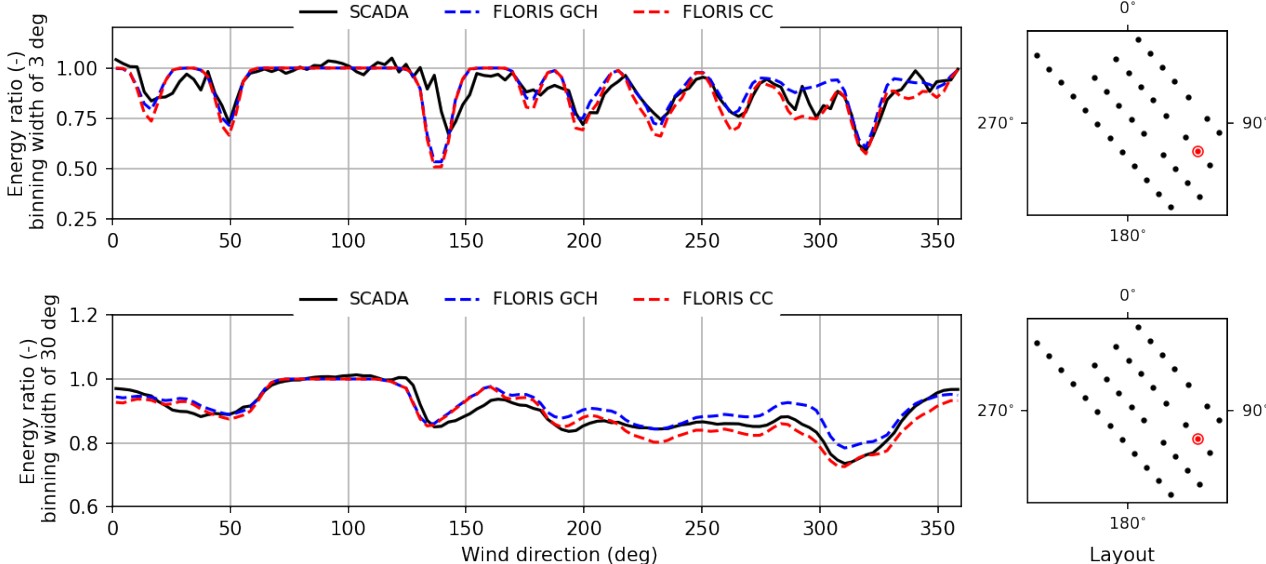

**Figure 15.** Energy ratio versus wind direction for a turbine located in the southeast region of the OWEZ wind farm. Note there is a lack of data for wind directions less than $150°$, potentially explaining the larger mismatch for the wake centered at $140°$. The reference turbines to which the energy ratio is normalized are the turbines that are both operating in freestream flow and within a 5 km radius of Turbine 22. The reference wind direction is determined by the nacelle headings from turbines directly neighboring Turbine 22.

## 5   Conclusions

This work presented the new cumulative-curl (CC) model, building on the previous Gauss-curl hybrid (GCH) model (King et al., 2020) and integrating near-wake calculations from Blondel and Cathelain (2020) and the cumulative wake effects deter-
mined by Bastankhah et al. (2021). The CC model increases accuracy in the wake and power prediction of larger wind farms, particularly in situations of many partial wakes and large downstream distances behind turbines, while maintaining the full-wake and small-farm accuracy of previous models. Model accuracy was demonstrated against both high-fidelity large-eddy simulations and historical data from physical wind plants. The CC model is able to accurately predict farm performance in both yawed and non-yawed operational strategies. Additionally, the CC model was implemented in a vectorized framework
to reduce the computational cost of calculating wake effects over many wind conditions. As such, the CC model now enables more reliable simulation studies for both small and large offshore wind farms, and with the low computational cost involved, the CC model makes an ideal candidate for wake-steering and layout optimization.

Future work includes adding additional aerodynamic effects not explicitly captured in the CC model. For example, wind farm blockage can occur in large wind farms. Large farms create a pressure field in front of the farm, most notably at the





turbines in the middle of the upstream row, that decreases the wind speed at those turbines compared to the ambient flow. This shows up as a heterogeneity in the inflow wind speed across the upstream row of turbines. Beyond calculating blockage, heterogeneous inflow is another effect to be captured. As mentioned, heterogeneous wind directions and wind speeds can occur from wind farm blockage but also from nearby wind farms and from coastal effects. Such heterogeneity effects are particularly noticeable for very large offshore wind farms such as the Anholt wind farm, which is over 10 km long in its dominant direction.

While the effects of blockage or neighboring farms are not inherently included in the CC model, the model does allow different ambient wind speeds to be simulated across the wind farm. Thus, if one can derive the subsequent effects of blockage, the CC model can include its effects. For this reason, heterogeneity and blockage effects are not considered an important model flaw currently, but rather a matter of appropriate model usage. Lastly, the consideration of flow acceleration, or speed-ups, which can occur around corners of wind farms and between rows where parallel wakes diffuse laterally and squeeze the flow, would

further improve the overall prediction capabilities of the CC model.



*Author contributions.*   All authors contributed to text. CJB implemented CC model in FLORIS and performed comparisons. PF performed analysis of SOWFA simulations and lead comparison to SCADA data. BD designed SCADA data comparisons. JK implemented GCH, Blondel, and cumulative wake models. MC lead the development of the WRF-driven SOWFA simulations and assisted with analysis. RM developed the FLORIS codes and models implementing CC.

*Competing interests.*   None

*Acknowledgements.*   The authors would like to thank Nicolai G. Nygaard, Sidse D. Hansen, and Peter Grønborg from Ørsted for the insightful discussions on the topics of data filtering and model validation, and for facilitating historical data from the Anholt and Westermost Rough offshore wind farms. The authors would like to express their gratitude to Jasper Kreeft and Nick Smith from Shell for facilitating historical data from the OWEZ offshore wind farm.

This work was authored by the National Renewable Energy Laboratory, operated by Alliance for Sustainable Energy, LLC, for the U.S. Department of Energy (DOE) under Contract No. DE-AC36-08GO28308. Funding provided by the National Offshore Wind Research and Development Consortium under Agreement FIA-19-16408-0. The views expressed in the article do not necessarily represent the views of the DOE or the U.S. Government. The U.S. Government retains and the publisher, by accepting the article for publication, acknowledges that the U.S. Government retains a nonexclusive, paid-up, irrevocable, worldwide license to publish or reproduce the published form of this work, or

allow others to do so, for U.S. Government purposes. A portion of this research was performed using computational resources sponsored by the Department of Energy's Office of Energy Efficiency and Renewable Energy and located at the National Renewable Energy Laboratory.



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
