# Peer review of "Addressing deep array effects and impacts to wake steering with the cumulative-curl wake model"

_Wind Energy Science, 2022_

## Author Comment (AC3)

**Response to Reviewer 1's comments:**

In this article, the authors develop a new analytical wind-farm flow model which accounts for larger wake losses deep in a wind farm and more persistent wake losses in the far-wake region. The model, which is named the cumulative-curl model, is then extensively validated against LES results (from SOWFA) and SCADA data for three different offshore wind farms. Two main conclusions can be drawn from the validation campaign. First, the model compares reasonably well with numerical data and measurements in all cases considered. Second, the new model outperforms the GCH one in cases with large wake losses and wake recovery over large turbine distances while it performs similarly in the other cases. I believe that this paper is of interest to the wind energy community as it shows the potential of a new analytical flow model which partially solves a long-standing problem, that is the mismatch in predictions in case of deep-array effects. Moreover, I enjoyed reading this work, which is well presented and well structured. Here below, you can find some scientific questions and technical comments.

The authors thank the reviewer for their kind comments.

Scientific comments/questions:

1. Abstract: it could be more descriptive. For instance, the authors write "Two points of model discrepancy were identified therein. The present article addresses those two concerns and presents the cumulative-curl (CC) model."Which are these two concerns? They will become clear once reading the text, but I would find it useful to mention them here. Also, would it be possible to translate the "improved accuracy" or "greatly reducing the computational time" into percentage and speed-up values?

The authors thank the reviewer for their feedback. Changes have been made to better explain the two modeling discrepancies identified by the recent GCH comparison paper, as shown below. For the second request, the authors don't have a non-vectorized form of the CC model to compare to for speed-up values, but the authors have seen speed-ups of multiple orders of magnitude for the default Gaussian wake model in FLORIS after vectorization. For a percentage increase in accuracy, the authors set out to show an overall improvement in the prediction capabilities of the CC model compared to the default Gaussian wake model. This is trend is established by the presented results, and the authors plan to perform a more detailed analysis of performance in future work.

**Abstract.** Wind farm design and analysis heavily rely on computationally efficient engineering models that are evaluated many times to find an optimal solution. A recent article compared the state-of-the-art Gauss-curl hybrid (GCH) model to historical data of three offshore wind farms. Two points of model discrepancy were identified therein: poor wake predictions for turbines experiencing a lot of wakes and wake interactions between two turbines over long distances. The present article addresses

5    those two concerns and presents the cumulative-curl (CC) model. Comparison of the CC model to high-fidelity simulation data and historical data of three offshore wind farms confirms the improved accuracy of the CC model over the GCH model in situations with large wake losses and wake recovery over large inter-turbine distances. Additionally, the CC model performs comparably to the GCH model for single- and fewer-turbine wake interactions, which were already accurately modeled. Lastly, the CC model has been implemented in a vectorized form, greatly reducing the computation time for many wind conditions.

10   The CC model now enables reliable simulation studies for both small and large offshore wind farms at a low computational cost, thereby making it an ideal candidate for wake-steering optimization and layout optimization.

2.  Section 2: I understand that it is more practical to cite others' works instead of re-writing the full model. Eventually, this also makes the article easier to read. However, I find it difficult to follow at times. For instance, the authors mention that the original cumulative wake model proposed in Bastankhah et al. (2021) does not include a near-wake model. However, in the current work, this deficiency is overcome by representing the near-wake region with a super-Gaussian wake model. How is this done? Is there an analytical derivation? I think that the reader would benefit from a more in-depth description of the model, which could also be provided in the appendix.

The authors appreciate the reviewer's feedback. The reviewer is correct in that a near wake model is added to the cumulative wake model. There is not an analytical derivation for this, but rather the combination is done in a heuristic manner that yields improved prediction results over the current default Gaussian wake model in FLORIS. The model description section has been reworked to better describe how the model is derived and its relevant components.

The wake deficit takes the super-Gaussian form proposed in Blondel and Cathelain (2020), as shown in Eq. (1).

$$\frac{\Delta u}{U_h} = C_n e^{\left(\frac{-\tilde{r}^m}{2\sigma_n^2}\right)} \tag{1}$$

The original cumulative wake model proposed in Bastankhah et al. (2021) does not include a near-wake model, which the super-Gaussian proposed by Blondel and Cathelain (2020) provides. $\Delta u$ is the wake velocity deficit, $U_h$ is the average wind velocity inflow across the current turbine rotor, $\sigma_n$ is the wake half-width, $n$ is the index of the current turbine, $\tilde{r}$ is the radial distance from the wake center normalized by the wind turbine diameter, and $m$ is the super-Gaussian order. $\sigma_n$, $\tilde{r}$, and $m$ are defined as:

$$\sigma_n = k\tilde{x} + \epsilon \tag{2}$$

$$\tilde{r} = \frac{\sqrt{(y - y_i - \delta_y)^2 + (z - z_i)^2}}{D} \tag{3}$$

$$m = a_f e^{(b_f \tilde{x})} + c_f \tag{4}$$

where $\tilde{x}$ is the downstream distance normalized by rotor diameter $\tilde{x} = \frac{|x - x_n|}{D}$, $i$ is the index of the upstream turbines, $y$ are the lateral locations, $z$ are the vertical locations, $\delta y$ is the lateral wake deflection, $k = a_s \text{TI} + b_s$ is the wake expansion parameter, TI is the turbulence intensity, and $a_s$, $b_s$, $a_f$, $b_f$, and $c_f$ are tuned model parameters. $\epsilon$ is defined as:

$$\epsilon = (c_{s1} C_{t,n} + c_{s2}) \cdot \sqrt{\beta} \tag{5}$$

$$\beta = 0.5 \cdot \frac{1.0 + \sqrt{1.0 - C_{t,n}}}{\sqrt{1.0 - C_{t,n}}} \tag{6}$$

where $C_t, n$ is the thrust coefficient of the current turbine, and $c_{s1}$ and $c_{s2}$ are both tuned model parameters.

The cumulative wake effect is included by adding the summation of $\sum_{i=1}^{n-1} \lambda_{ni} C_i / U_o$ into the wake center velocity deficit $C_n$ of the current turbine $n$, shown in Eq. (7).

$$C_n = \left(1 - \sum_{i=1}^{n-1} \lambda_{ni} \frac{C_i}{U_o}\right) \left(a_1 - \sqrt{a_2 - \frac{mC_{t,n} \cos(\gamma_n)}{16.0\Gamma(2/m)\sigma_n^{4/m}\left(1 - \sum_{i=1}^{n-1} \lambda_{ni} \frac{C_i}{U_o}\right)^2}}\right) \tag{7}$$

$\lambda_{ni}$ captures the wake contribution of upstream wind turbine $i$ on the value of $C_n$, as described in Bastankhah et al. (2021). Here, $U_o$ is the freestream velocity, $\gamma_n$ is the current turbine's yaw angle, and $\Gamma$ is the gamma function. $\lambda_{ni}$, $a_1$, and $a_2$ are defined as:

$$\lambda_{ni} = \frac{\sigma_n^2}{\sigma_n^2 + \sigma_i^2} e^{-\frac{(y_n - y_i - \delta y)^2}{2(\sigma_n^2 + \sigma_i^2)}} e^{-\frac{(z_n - z_i)^2}{2(\sigma_n^2 + \sigma_i^2)}} \tag{8}$$

$$a_1 = 2^{(2/m-1)} \tag{9}$$

$$a_2 = 2^{(4/m-2)} \tag{10}$$

The same deflection model is used from the GCH model, which includes secondary steering due to yawed wakes, and the yaw added recovery effects are also captured when updating the local turbulence intensity (TI) conditions for the current turbine, as defined in King et al. (2020).

3. Section 3.1 (line 167): Which is the horizontal resolution used in the precursor simulation? Also, for how long the precursor simulations have been advanced in time?

The authors thank the reviewer for their questions. The resolution in the horizontal and vertical directions is 10 meters. Text has been added to clarify this. Also, the precursor simulations were run for 20000 seconds, which has also been added to the manuscript.
* * *
165  NREL OpenFAST wind turbine structural-aerodynamics-servodynamic simulator (202).

Simulations are run in two stages. In the first stage, often termed a "precursor", the computational domain (roughly 10 km × 5 km horizontally and 3 km tall with 10 m resolution in each direction within the boundary layer) is laterally periodic and the flow is allowed to cycle through the domain for 21600 seconds (6 hours of simulation time), during which a realistic
* * *
4. Section 3.1 (line 197): From where does this formula come? Consider adding a reference. Also, I would consider a different notation for the wind speed (ws can be seen as w*s in a mathematical expression).

The authors appreciate the reviewer's feedback. The coefficient in equation 13 comes from experiments at NREL. This, along with the wind speed variable, has been updated in the text as shown below.
* * *
$$TI = \sqrt{TKE * 1.07}/w_{spd} \qquad (13)$$

where $TKE$ is turbulent kinenetic energy and $w_{spd}$ is the wind speed. The coefficient 1.07 is based on experiments in homo-
200  geneous shear flow performed at NREL.
* * *
5. Section 3.1 (line 230): For how long the wind-farm simulations are run in SOFWA? In the article, it is reported only the spin-up time (1200 s), but not the time over which statistics are collected.

The authors thank the reviewer for pointing out this missing information. The length of simulation used for the time-averaging was 2400 seconds, and this information has been added to the manuscript.
* * *
Full wind farm simulations are run for a range of wind turbine yaw angles and lateral turbine locations, the latter to provide more realizations of the same flow for more converged flow statistics. In further analysis, the first 1,200 s of the simulation
230  results are omitted because they relate to wake development and start-up effects. The remaining simulation output is time-averaged over 2400 s to obtain a steady-state representation of the wind turbine and farm performance. Using the time-averaged
* * *
6. Section 3.2: Over which region the velocity is spatially averaged to produce the power trends observed in Fig. 1? Is the height-dependent velocity profile given by the precursor simulations taken into account in the analytical models (GCH and CC)? Or do the analytical model assumes a uniform inflow velocity profile? If

so, how is the velocity magnitude estimated? Does it refer to the velocity at hub height? This information can improve the interpretation of the validation results.

The authors thank the reviewer for their questions. Both in SOWFA and FLORIS, the region over which the velocity is spatially averaged is the circular region of the rotor to determine the effective rotor velocity at hub height. This includes all the points that fall within the rotor area. For the analytical models, they assume a log-law application of shear to define the inflow velocity profile. A general shear component of 0.12 was assumed based on previous work (this is the default value currently in FLORIS) and analysis was not done to match the exact shear profiles to the SOWFA data. Text has been added to clarify this in the paper.

an upstream turbine. To determine the effective rotor velocity at the different locations downstream, an average velocity of the points within the hypothetical downstream turbine's rotor area is computed for both SOWFA and the analytical models. In these simulations, the upstream turbine is aligned with the flow and thus wake steering is not yet considered. The TI assumed in
245 the GCH and CC models was selected to yield perfect agreement with SOWFA at a $7D$ distance downstream. This method was chosen to ensure good wake deficit predictions at a distance downstream that is similar to relative distances between turbines in offshore wind farms due to the significant impact that TI can have on analytical models. Also, the previous Gaussian model's predictions had trouble giving good predictions on either side of this downstream distance $7D$ when tuned, for example at $5D$ and $10D$. The $7D$ value was chosen to show that the new model's predictions are improved overall at other distances
250 compared to the old model. A log-law approximation of shear was applied to the background inflow using the default settings in FLORIS to approximate the shear that develops in the SOWFA simulations. Proceeding downstream from $7D$, Fig. 1 shows

7. Section 3.2 (line 243): The authors mention that "The TI assumed in the GCH and CC models was selected to yield perfect agreement with SOWFA at a 7Ddistance downstream." I believe that if the TI value was tuned so that a zero error would occur at 5D or 10D, the validation would look different. Therefore, why not use the ambient TI reported in table 1 (i.e. the ambient TI values given by the precursor simulations)? Please, comment on this.

The authors thank the reviewer for their comments. The TIs reported in Table 1 are estimated using equation 13 from the WRF data and give a general sense of the turbulence intensity for the precursors. However, as the flow in the precursors develop, this exact TI level cannot be ensured, and thus those values in Table 1 are just estimates used to down select from the different SCADA/precursor datasets, as described in Section 3.1. For the analytical models, the TI was tuned to give zero error in the hypothetical turbine power production at a distance downstream of 7D as this is a common relative distance between turbines for offshore wind farms. 7D was also chosen to show that the old model had difficulty when tuned at 7D to match at 5D and 10D, while the new model does better through this range of distances and was the point the authors were attempting to make with this process. Also, due to the impact of TI on the analytical models, the authors chose this method to give better wake deficit

predictions, and thus better turbine power predictions over using the estimated TI from the WRF data that may not match exactly the turbulence that developed within the SOWFA precursors. Additional description has been added to the text to clarify the choice for using this method, as shown below.

245    the GCH and CC models was selected to yield perfect agreement with SOWFA at a $7D$ distance downstream. This method was
chosen to ensure good wake deficit predictions at a distance downstream that is similar to relative distances between turbines in
offshore wind farms due to the significant impact that TI can have on analytical models. Also, the previous Gaussian model's
predictions had trouble giving good predictions on either side of this downstream distance $7D$ when tuned, for example at
$5D$ and $10D$. The $7D$ value was chosen to show that the new model's predictions are improved overall at other distances
250    compared to the old model. A log-law approximation of shear was applied to the background inflow using the default settings

8. Section 3.2.2: How is the ambient TI evaluated in this section? Are the authors using the ambient TI shown in table 1? I'm concerned about the ambient TI because usually, the analytical wake model predictions are strongly dependent on this value. Moreover, it is important to mention the ambient TI in case of the reader would like to reproduce some of the results. Note that the same question holds also for sections 3.3 and 3.4. Finally, it is not clear to me how the added TI is computed. Would it be possible to include this in the text?

The authors thank the reviewer for their questions. The ambient TIs used in sections 3.3 and 3.4 are the same that were determined in section 3.2, as described in the response immediately above. The wake-added TI is computed using the default wake-added TI model in FLORIS, which is the Crespo-Hernandez model. This detail has been added to the text, as shown below.

**Table 2.** Summary of the tuned turbulence intensity (TI) values used in the FLORIS simulations.

| Precursor | Wind Farm | TI (%) |
|-----------|-----------|--------|
| ah_3 | Anholt | 8.8% |
| ah_4 | Anholt | 9.3% |
| ah_5 | Anholt | 9.5% |
| ha_1 | Hawaii | 10.5% |
| ha_2 | Hawaii | 14.3% |
| hb_4 | Humboldt | 8.3% |

compared to the old model. The default wake-added turbulence model in FLORIS, Crespo-Hernandez, is used for the FLORIS
simulations. A log-law approximation of shear was applied to the background inflow using the default settings in FLORIS to

9. Figure 2: I find this plot difficult to read. Have you tried using different symbols instead of different lines for the various precursor cases? That is just a suggestion since it may make it worst.

We thank the reviewer for their feedback. We have tried using different symbols for the various cases (as opposed to the different lines currently used), and it does make it more difficult to interpret, as the reviewer suggested may happen. For this reason, we feel the current formatting is the best way to display the data.

10. Figure 5: Why the GCH and CC first-row turbine power is lower than the one predicted by SOFWA in all cases? This mismatch (although limited to a few percentage points) could lead to a bias in the measurements further downstream. Moreover, I would find it very interesting to include the prediction of, for instance, the Jensen model in the current figure. This could highlight how much better the models have become at matching LES results. However, I also understand that this could be out of the scope of the current manuscript.

The authors thank the reviewer for their questions. The slight differences in power of the lead turbine between the SOWFA simulations and the FLORIS predictions is most likely due to the difference in flow in SOWFA that exists laterally. The wind speeds for the FLORIS simulations were initially tuned to single turbine powers in SOWFA, but for Figure 7, the averaged power from a row of turbines was used from the Reference Farm layout. So the difference in power most likely stems from the variations in flow laterally in the SOWFA simulation when they are averaged. The authors felt that keeping the original tuning was the best route forward for the larger farm comparisons.

Also, while it would be interesting to compare the Jensen model to the Gaussian wake models displayed here, the authors feel that is also outside the scope of this research effort which focuses on improving Gaussian wake models. A broader model comparison effort will be considered for future work.

11. Figure 7: Which are the yaw angles applied at every turbine row? The authors mention that "This pattern assumes a large yaw misalignment angle for the first row of turbines, which then decreases linearly to zero for the last row of wind turbines.". However, they do not provide the yaw misalignment for the first row of turbines. This information is necessary in case of the reader would like to reproduce the results.

The authors thank the reviewer for pointing out the missing yaw angles. This information has been added to the text as shown below.

325    of wind turbines. Specifically, the first turbine is yawed 30 degrees, and then each turbine is yawed 5 degrees less, down to 0 degrees for the last turbine Due to high-computing facility resource limitations, these simulations were not performed for the Rotated and Gap farms. Figure 7 shows that the GCH model has a tendency to overestimate the wake losses at downstream

**Figure 7.** Comparing the ratio of the average power produced per wind turbine row (average indicated by the point, shaded band indicating confidence interval) for the Reference Farm type across precursors when applying a pattern of decreasing yaw angles per row starting at 30 deg and going to 0 deg by increments of 5 deg.

12. Section 4: Very nice and strong validation of the model. I have only a minor question here. In figure 15 (top panel), both models predict a lower energy ratio for a wind direction of 140 degrees than 320 degrees. Why is this happening? In fact, in the first case, turbine 22 operates in the wake of one turbine while in the second case it operates in the wake of six upwind turbines.

The authors thank the reviewer for their feedback. For the SCADA data used in Figure 15, after quality control, there was less data available for wind directions less than 150 degrees, which potentially explains the larger mismatch that the reviewer has pointed out at 140 degrees. This is further supported by the bottom plot in Figure 15 where a larger binning width of 30 degrees is used and the resulting prediction better matches that of the data. A comment on this discrepancy is already included in the caption for Figure 15, so no changes have been made.

13. I noticed that in many cases the caption of the figures also contains interpretations of the results. I would use the caption only for describing the figure, therefore moving the interpretation of the results in the main text.

The authors appreciate the reviewer's comments. However, it is our viewpoint that descriptive captions for the Figures aid in the reader's ability to quickly scan the manuscript and gain some initial insight into the results. As such, the authors respectfully will keep the captions as-is.

Technical comments:

1. Line 247: Typo "largely largely"

Thank you for catching this typo. This has been corrected.

2. line 320: re-phrase the sentence

The authors thank the reviewer for catching this mistake. The sentence has been corrected, as shown below.

> model much better represents wake recovery under partial wake overlap and in irregularly spaced wind farms. When applying
> a pattern of decreasing yaw angles per row, the CC model seems to under-predict the power lift from wake steering compared
> 330    to the SOWFA data. This wake steering strategy and its effects on the CC model will be considered in future work.

**Response to Reviewer 2's comments:**

This paper gives adds the super-Gaussian model of Blondel and Cathelain and the cumulative wake superposition of Bastankhah et al. to the Gauss-curl hybrid model to address issues with deep array effects. The comparisons to high-fidelity models and field data are comprehensive. The paper is a worthwhile addition to the considerable research on wind farm wake modeling that is necessary for wind farm design and control.

The authors are thankful for the reviewer's comments.

Introduction: I enjoyed this clear discussion of the complications of wake modeling (including wake super position and near and far wake models) in various wind farm configurations. Two issues that could use some discussion are (1) momentum conserving models and linearized momentum conserving models (often called mass conserving models) and (2) the choice of wake expansion rate through turbulence characteristics.

The authors thank the reviewers for their suggestions. The cumulative wake model proposed by Bastankhah fits in the first category suggested by the reviewer. As it is relevant to the paper at hand, some discussion of this model has been added to the introduction. However, the authors have decided not to add additional discussion regarding the choice of wake expansion rate through turbulence characteristics as we feel this is too far from the current scope of this effort. The added discussion is shown below.

> These superposition methods, while proposed to conserve momentum under certain assumptions, are not based in theoretical
> 60    derivations and are more appropriately classified as empirical relationships, as discussed in Zong and Porté-Agel (2020). A mo-
> mentum conserving superposition method is derived and detailed in Zong and Porté-Agel (2020), showing good improvement
> over the other methods available in literature. More recently, Bastankhah et al. (2021) proposed an analytical solution based on
> the mass and momentum conservation principal that implicitly includes wake superposition in the calculation of the velocity
> deficit. This model is known as the cumulative wake model, and in Bastankhah et al. (2021) is shown to have improvement for
> 65    larger wind farm predictions.

Section 2.2: It is hard to decipher where each of these equations come from and how they have been modified in this implementation. Is there a consistent theoretical basis for adding the Blondel & Cathelain model and cumulative model of Bastankhah et al. to the GCH model? Or are the additions heuristic?

The authors appreciate the reviewer's feedback. The additions of the Blondel & Cathelain model to the cumulative Bastankhah model are heuristic. The insertion of the near-wake model into the cumulative wake model, along with tuning, was found to give slower wake recovery over distance downstream, which is what is reflected in the SOWFA simulations and in feedback from industry partners. A theoretical basis could potentially be derived but is left for future work. the equations and their origins have been restructured, as shown below, to better illustrate the derivation of the model.

The wake deficit takes the super-Gaussian form proposed in Blondel and Cathelain (2020), as shown in Eq. (1).

$$\frac{\Delta u}{U_h} = C_n e^{\left(\frac{-\tilde{r}^m}{2\sigma_n^2}\right)} \tag{1}$$

The original cumulative wake model proposed in Bastankhah et al. (2021) does not include a near-wake model, which the super-Gaussian proposed by Blondel and Cathelain (2020) provides. $\Delta u$ is the wake velocity deficit, $U_h$ is the average wind velocity inflow across the current turbine rotor, $\sigma_n$ is the wake half-width, $n$ is the index of the current turbine, $\tilde{r}$ is the radial distance from the wake center normalized by the wind turbine diameter, and $m$ is the super-Gaussian order. $\sigma_n$, $\tilde{r}$, and $m$ are defined as:

$$\sigma_n = k\tilde{x} + \epsilon \tag{2}$$

$$\tilde{r} = \frac{\sqrt{(y - y_i - \delta_y)^2 + (z - z_i)^2}}{D} \tag{3}$$

$$m = a_f e^{(b_f \tilde{x})} + c_f \tag{4}$$

where $\tilde{x}$ is the downstream distance normalized by rotor diameter $\tilde{x} = \frac{|x - x_n|}{D}$, $i$ is the index of the upstream turbines, $y$ are the lateral locations, $z$ are the vertical locations, $\delta y$ is the lateral wake deflection, $k = a_s\text{TI} + b_s$ is the wake expansion parameter, TI is the turbulence intensity, and $a_s$, $b_s$, $a_f$, $b_f$, and $c_f$ are tuned model parameters. $\epsilon$ is defined as:

$$\epsilon = (c_{s1}C_{t,n} + c_{s2}) \cdot \sqrt{\beta} \tag{5}$$

$$\beta = 0.5 \cdot \frac{1.0 + \sqrt{1.0 - C_{t,n}}}{\sqrt{1.0 - C_{t,n}}} \tag{6}$$

where $C_t, n$ is the thrust coefficient of the current turbine, and $c_{s1}$ and $c_{s2}$ are both tuned model parameters.
* * *
The cumulative wake effect is included by adding the summation of $\sum_{i=1}^{n-1} \lambda_{ni}C_i/U_o$ into the wake center velocity deficit $C_n$ of the current turbine $n$, shown in Eq. (7).

$$C_n = \left(1 - \sum_{i=1}^{n-1} \lambda_{ni}\frac{C_i}{U_o}\right)\left(a_1 - \sqrt{a_2 - \frac{mC_{t,n}cos(\gamma_n)}{16.0\Gamma(2/m)\sigma_n^{4/m}\left(1 - \sum_{i=1}^{n-1}\lambda_{ni}\frac{C_i}{U_o}\right)^2}}\right) \tag{7}$$

$\lambda_{ni}$ captures the wake contribution of upstream wind turbine $i$ on the value of $C_n$, as described in Bastankhah et al. (2021). Here, $U_o$ is the freestream velocity, $\gamma_n$ is the current turbine's yaw angle, and $\Gamma$ is the gamma function. $\lambda_{ni}$, $a_1$, and $a_2$ are defined as:

$$\lambda_{ni} = \frac{\sigma_n^2}{\sigma_n^2 + \sigma_i^2}e^{-\frac{(y_n - y_i - \delta y)^2}{2(\sigma_n^2 + \sigma_i^2)}}e^{-\frac{(z_n - z_i)^2}{2(\sigma_n^2 + \sigma_i^2)}} \tag{8}$$

$$a_1 = 2^{(2/m - 1)} \tag{9}$$

$$a_2 = 2^{(4/m - 2)} \tag{10}$$

The same deflection model is used from the GCH model, which includes secondary steering due to yawed wakes, and the yaw added recovery effects are also captured when updating the local turbulence intensity (TI) conditions for the current turbine, as defined in King et al. (2020).

Section 2.2: This model has a large number of free parameters, which makes it more difficult to use. Could you discuss in more depth how these parameters are selected to make the model more widely useable.

The author thanks the reviewer for their feedback. For this effort, the tuned model parameters were not changed from their default values which were determined by Cathelain et al. Text has been added to the manuscript to refer readers to the proper source for more information on the tuning of these parameters.

TI is the turbulence intensity, and $a_s$, $b_s$, $a_f$, $b_f$, and $c_f$ are tuned model parameters (for details on tuning the parameters,
115 see Cathelain et al. (2020)). $\epsilon$ is defined as:

where $C_{t,n}$ is the thrust coefficient of the current turbine, and $c_{s1}$ and $c_{s2}$ are both tuned model parameters (for details on tuning the parameters, see Cathelain et al. (2020)).
120    The cumulative wake effect is included by adding the summation of $\sum_{i=1}^{n-1} \lambda_{ni} C_i / U_o$ into the wake center velocity deficit

All graphs: Please use vector formats for these images and use consistent font sizes. The resolution is fairly low and the font is sometimes hard to read. Use more descriptive titles without using underscores.

The authors thank the reviewer for their feedback. Many of the relevant plots have been updated throughout the manuscript, with examples below.

[Figure]

**Figure 1.** Power production of a downstream wind turbine located at various distances behind a yaw-aligned upstream wind turbine for

[Figure]

**Figure 3.** The power production of a hypothetical turbine 7 rotor diameters distance from an upstream wind turbine. The upstream wind

[Figure]

**Figure 5.** Comparing the power produced by row across farm types and models. Note that the plot points are average power for a row (with

Figure 1: The improvement here is not as apparent to me as claimed in the text. I would have assumed that the super-Gaussian near wake model would improve the agreement in the near wake. In fact, the opposite seems to be the case. Furthermore, the choice to tune the results at x/D=7 affects the model accuracy. If the tuning had been done at x/D=3 the results might be quite different. A better approach would be to minimize the error over all measurements.

The authors appreciate the reviewer's feedback. We have updated the text to reflect that the major prediction improvements occur not in the near wake but in the medium to far wake regions. There is existing discussion in the paper that covers the significant impact of the near-wake model on the development of the wake in the far wake region, where the authors were looking for major improvement. Additional improvement of the near wake region predictions is left for future work.

> wake recovery in the GCH model is overestimated with respect to SOWFA, while the new CC model shows better agreement. Note that there is still an error in the very near wake region, but the CC model matches much better in the medium to far wake regions where other turbines will most likely exist. The focus of this effort was to specifically improve the accuracy of the
> 255    wake model in the far wake region, and additional improvement of the near wake deficit prediction is left for future work. This trend of slower wake recovery at large downstream distances is supported in literature (Nygaard et al., 2020).

For the analytical models, the TI was tuned to give zero error in the hypothetical turbine power production at a distance downstream of 7D as this is a common relative distance between turbines for offshore wind farms. 7D was also chosen to show that the old model had difficulty when tuned at 7D to match at 5D and 10D, while the new model does better through this range of distances and was the point the authors were attempting to make with this process. Updates to the text with further explanation of this are shown below.

> 245    the GCH and CC models was selected to yield perfect agreement with SOWFA at a $7D$ distance downstream. This method was chosen to ensure good wake deficit predictions at a distance downstream that is similar to relative distances between turbines in offshore wind farms due to the significant impact that TI can have on analytical models. Also, the previous Gaussian model's predictions had trouble giving good predictions on either side of this downstream distance $7D$ when tuned, for example at $5D$ and $10D$. The $7D$ value was chosen to show that the new model's predictions are improved overall at other distances
> 250    compared to the old model. A log-law approximation of shear was applied to the background inflow using the default settings

Section 3.2.2: Since these results are for a single turbine, they are only including the effect of including the Blondel & Cathelain model in the GCH model and the Bastankhah model does that have an impact, correct? Or am I misunderstanding that? I suggest adding some discussion on what aspects of the model are being tested here.

The authors thank the reviewer for their questions. The results in Section 3.2.2 are looking at the power production of a hypothetical turbine that is swept laterally behind

a real, yawed turbine. This lateral sweeping occurs at a downstream distance of 7 rotor diameters. What this is testing is the implementation of the GCH effects onto the blended Bastankhah/Blondel & Cathelain model. Because of the good agreement shown in Figure 3, the authors are confident that the GCH models have been implemented correctly around the blended cumulative wake model. The simulation is already described in section 3.2.2, so detail has been added to confirm the purpose of this analysis, as shown below.
* * *
agreement in wake-steering experiments (Fleming et al., 2021). This confirms that the GCH modeling aspects were correctly integrated into the CC model.

**3.3 Farm-scale wake effects**
* * *
Section 4: The paper has a lot of great data for the comparison. While the graphs are very instructive to understand the differences between the models, it's hard to compare the average error. Could you provide average error results for each of these graphs in a table?

The authors appreciate the reviewer's feedback. While it is possible to produce the average error results, the authors feel including the values at this point would detract from the main purpose of the paper, which was to show that by including a near wake model with the cumulative wake model, we can improve the overall deeper wake prediction when compared to our current Gaussian wake model. These trends are proved out in the included plots, and we plan to improve this matching through various methods in future work.

Figure 10: What do the color plots on the right represent? There is no label and they are difficult to read.

The authors appreciate the reviewer's question. The color plots on the right are just meant to act as a supplement to the main plots on the left, showing the overall farm layout and the section of the farm as well as the turbines that are being examined. The figure has been made larger in the LaTeX document and text has been added to the caption to explain the plots on the right, as shown below.

[Figure]

**Figure 10.** Ratio of energy produced by wind turbines in a row with respect to the energy produced by the first turbine in the row. The energy ratios are normalized to the most upstream wind turbine in the row. The wind direction used to bin the data is assumed to be equal to the nacelle heading of the most upstream wind turbine. The four turbine arrays considered are all from the Anholt wind farm, which is an excellent candidate for these validation studies because of its sheer size (111 wind turbines). The layout of the Anholt farm is shown on the right of each plot, with the turbine IDs highlighted for each case.

Sections 3&4: I suggest changing the "Reflection" subsections to "Discussion."

The authors thank the reviewer for their suggestion. The changes have been made in the manuscript.

**3.5   Discussion**

**4.4   Discussion**

---

## Author Comment (AC4)

**Response to Reviewer 1's comments:**

The revised manuscript is well presented and well structured. The authors' response covers all doubts and questions raised, and appropriate changes have been applied to the manuscript. However, I still have some minor scientific and technical comments which you can find here below.

The authors thank the reviewer for their kind comments and for their assistance in improving the manuscript.

Scientific comments/questions

1. Line 244: in the current work, the wind-farm start-up phase lasts for 20 minutes. Have you checked whether the selected time horizon suffices for reaching a fully-developed statistically steady-state flow in and around the farm? I believe that this is important since the SOWFA outputs are further compared against steady-state models.

The authors thank for the reviewer for their comment. We have not done statistical analysis on the fully-developed SOWFA flow, but chose the length of start-up time based on previous experience and an approximate calculation that is based on how long it would take a flow at freestream velocity to travel through the full domain. In this case, for a 10 km long domain, at 8.5 m/s freestream (just below all the average wind speeds of the precursors), it would take wakes 10 km / 8.5 m/s ≈ 1200 seconds to reach the end of the domain. Based on this, and the fact the precursors were run for 21,600 seconds to establish a steady-state freestream, the authors felt confident in taking an average over 2400 seconds of simulation to capture pseudo-steady-state turbine/flow data. Additional explanation has been added to the text, as shown below.

> Full wind farm simulations are run for a range of wind turbine yaw angles and lateral turbine locations, the latter to provide more realizations of the same flow for more converged flow statistics. In further analysis, the first  1200 s of the simulation results are omitted because they relate to wake development and start-up effects. This value was chosen based on the calculation
> 245 of approximately how long it would take the wake of a wind turbine to propagate at freestream velocity (approximately 8.5 m/s) through the entire domain (10km streamwise). The remaining simulation output is time-averaged over 2400 s to obtain a steady-state representation of the wind turbine and farm performance. Using the time-averaged cubed wind speed field (as

2. Line 264: Are the coefficients of the Crespo-Hernandez wake-added turbulence model the original ones? If not, I would suggest including them in the manuscript (for instance, as done in Doekemeijer et al. Table A1).

The authors appreciate the reviewer's feedback. The Crespo-Hernandez model coefficients used are the same as are defined in the example input file for the Cumulative Curl model in the FLORIS GitHub repository. For clarity and reference, those values have been added as shown below.

265  $5D$ and $10D$. The $7D$ value was chosen to show that the new model's predictions are improved overall at other distances compared to the old model. The default wake-added turbulence model in FLORIS, Crespo-Hernandez, is used for the FLORIS simulations, with tuned parameter values as defined in the cumulative-curl input file in the FLORIS examples folders. For reference, those values are $ti\_initital = 0.01$, $ti\_constant = 0.9$, $ti\_ai = 0.83$, and $ti\_downstream = -0.25$. A log-law approximation of shear was applied to the background inflow using the default settings in FLORIS to approximate the shear

Technical comments

1. Throughout the manuscript, the precursor simulations are labelled with wind-farm names (see Tables 1 and 2). However, these precursor simulations are then used to drive the flow across idealized farm layouts (i.e. reference, rotated and gap farm). This creates confusion, especially while looking at Figs. 5, 6 and 7, where in the title of each subplot there are two wind-farm names. Hence, I would suggest labelling the precursor simulations differently.

The authors thank the reviewer for their feedback. Figures 1, 5, 6, and 7 have been more clearly labeled to indicate what precursor and layout are being used, as shown below.

[Figure]

[Figure]

[Figure]

[Figure]

2.

There are no references to Table 1 throughout the manuscript. This might be a typo.

The authors thank the reviewer for pointing this out. We have corrected the table references as shown below. Note that latexdiff did not pickup the update of the table number.

Table 1 presents a subset of the 23 precursor simulations that were performed. In the table TI represents the ambient turbu-
215  lence intensity as estimated via equation 11 using the TKE from WRF/NEWA. WD STD represents the standard deviation of
the ambient wind direction impinging the wind farm in degrees. Note that wind direction is not exactly 270° (left to right) at the

240  the precursors, the wind direction can vary slightly at heights other than 120 m. The Wind Direction column in Table 1 captures
this variation at 90 m (the hub height of the NREL 5 MW turbine), and is accounted for in the simulations.

**3.2  Single wake analysis**

The first set of wind farm simulations in SOWFA analyze the wake of a single turbine. For each SOWFA precursor simulation in Table 1, wind farm simulations of a single NREL 5 MW reference turbine are run (Jonkman et al., 2009).

255  **3.2.1  Wake recovery**

3.
   Line 179: typo a -> as

The authors thank the reviewer for pointing this out. The typo has been corrected.

Simulations are run in two stages. In the first stage, often termed a as "precursor", the computational domain (roughly 10

180  km × 5 km horizontally and 3 km tall with 10 m resolution in each direction within the boundary layer) is laterally periodic